

# Spatio-temporal structure of Baltic free sea level oscillations in barotropic and baroclinic conditions from hydrodynamic modelling.

Eugeny A. Zakharchuk[1,2], Natalia Tikhonova[1,2], Elena Zakharova[3,4]

[1]Saint-Petersburg University, Saint-Petersburg, 199004, Russia
[2] State Oceanographic Institute. Saint-Petersburg Branch, Saint-Petersburg, 199397, Russia
[3]Water Problem Institute, RAS, Moscow, 117971, Russia
[4]EOLA, Toulouse, 31400, France

*Correspondence to: Elena Zakharova (zavocado@gmail.com)*

**Abstract.** Free sea level oscillations in barotropic and baroclinic conditions were examined using numerical experiments based on a 3D hydrodynamic model of the Baltic Sea. In a barotropic environment, the highest amplitudes of free sea level oscillations are observed in the northern Gulf of Bothnia, eastern Gulf of Finland, and south-western Baltic Sea. In these areas, the maximum variance appears within the frequency range corresponding to periods of 13–44 hr. In a stratified environment, after the cessation of meteorological forcing, water masses relax to the equilibrium state in the form of mesoscale oscillations at the same frequencies as well as in the form of rapidly decaying low-frequency (seasonal) oscillations. The total amplitudes of free baroclinic perturbations are significantly larger than those of barotropic perturbations, reaching 15–17 cm. Contrary to barotropic, oscillations in baroclinic conditions are strongly pronounced in the deep-water areas of the Baltic Sea Proper. Specific spatial patterns of amplitudes and phases of free barotropic and baroclinic sea level oscillations identified them as progressive-standing waves representing barotropic or baroclinic modes of gravity waves and topographic Rossby waves.

## 1 Introduction

Free sea level oscillations are directly related to the eigenoscillations of sea basins. The spectral structure of eigenoscillations depends on sea basin scales, basin bathymetry, and land configuration. In eigenoscillation frequencies, the basin water masses return to equilibrium conditions after meteorological forcing (Lisitzin, 1974; Fennel end Seifert, 2008). Within these frequencies, the free oscillations resonate with wind forcing, resulting in an anomalous sea level rise followed by the inundation of coastal areas (Jönsson et al., 2008; Kulikov and Medvedev, 2013; Zakharchuk and Tikhonova, 2011). The investigation into free oscillations of sea basins is essential for correctly interpreting the spatio-temporal variability of physical, hydrochemical, and biological parameters, as well as for identifying the mechanisms responsible for this variability.

The Baltic Sea free oscillations are usually related to seiches. Seiches are free sea level fluctuations in an enclosed or semi-isolated basin, which occur as standing waves generated by external forcing and continue due to inertia after cessation of the initial force (Proudman, 1953; Lisitzin,1959; Pugh, 1987).

Previous studies based on spectral analysis of the Baltic Sea tide gauge records have described several Baltic seiche systems. One system is located on the West Baltic–Gulf of Finland axis and is characterised by periods of 26–32 h in the primary mode and 17–20 h in the secondary mode (Neumann, 1941; Magaard and Krauss, 1966;



Lisitzin, 1959, 1974; Kulikov and Medvedev, 2013). The primary mode of the second rapidly damping seiche
system situated in the Western Baltic- the Bothnia Bay axis, has 39 h period (Neumann, 1941).
Using a one-dimensional simplified numeric model, on the axis the Gulf of Finland–Danish Straits, Newman
(1941) detected seiches with a 27-h period. The amplitude of these seiches was usually less than 10 cm, rarely
reaching 40 cm. Nevertheless, higher amplitudes were not excluded.
Metzner et al. (2000) demonstrated that the Baltic free sea-level oscillations can be studied using satellite
altimetry combined with numerical modelling and in situ observations.
Research on free sea level oscillations based on numerical modelling found a more complex system of seiches in
the Baltic Sea. Using a two-dimensional shallow water model with 10 km spatial resolution, Wübber and Krauss
(1979) suggested ten modes of the Baltic Sea eigenoscillations. The first four modes have periods of 31, 26, 22,
and 20 h, respectively. The authors noted that the eigenoscillations were significantly modified by the Coriolis
force. Earth's rotation transforms all modes of eigenoscillations to positive amphidromic waves. As a result, the
period of oscillations may diminish (if this period is higher than an inertial period) or may increase (if it is lower
than the inertial period).
Subsequently, Jonsson et al. (2008), based on the analysis of linear shallow-water model simulations, identified
three different local oscillatory modes: in the Gulf of Finland (with two 23 and 27 h periods), Danish Belt Sea
(with periods of 23–27 h), and Gulf of Riga (with 17 h periods). The authors attributed these variations to
seiches and noted that they were not connected to each other. However, this conclusion is not convincing, as it
was not supported by the spatio-temporal distribution of the oscillation phases. The authors also suggested that
the Baltic free sea level oscillations can be related to Kelvin waves that propagate from the Gulf of Finland into
the Baltic Proper along the coastline.
Another study (Zakharchuk et al., 2004) that was based on simulation results of a hydrodynamic three-
dimensional model implies that low-frequency free oscillations in the Baltic Sea represent the topographic
Rossby waves because their phase velocity is significantly lower than that of the barotropic gravity waves (GW).
All previous studies based on numerical modelling investigated only the barotropic variations in the Baltic sea
level, while an actual sea basin is a baroclinic system. The specifics of the relaxation of the Baltic Sea water
masses to equilibrium after the cessation of anemobaric forces in baroclinic conditions remain unclear.
The present study investigates the difference between barotropic and baroclinic free sea level oscillations in the
Baltic Sea using a three-dimensional hydrodynamic model. First, the capability of the model to simulate sea
level fluctuations in different parts of the Baltic Sea was verified against in situ tide gauge observations (Section
2.3). Then, the spatio-temporal structure of the sea level variations in barotropic (Section 3.1) and baroclinic
(Section 3.2) conditions is analysed using Fourier analyses of the model outputs. To interpret the detected free-
sea level oscillations we compared the estimated phase speed of the modelled oscillations with the theoretical
phase speed values of barotropic and baroclinic gravity waves and discuss the results in Section 4.

**2  Data and methods**

A three-dimensional non-linear baroclinic model developed by the Institute of Numerical Mathematics of the
Russian Academy of Science (Institute of Numerical Mathematics Ocean Model or INMOM) was selected for
studying the Baltic free sea level oscillations (Diansky et al., 2006; Zalesny et al., 2012). The model was



configured for the Baltic Sea basin and run in its basic setup to ensure the credibility of the sea level simulations.
Then, the model was re-configured for two numerical experiments to represent the barotropic and baroclinic
conditions in the Baltic Sea.

**2.1 Model description**

The INMOM is based on primitive equations of ocean hydrodynamics in spherical coordinates and on
hydrostatic and Boussinesq approximations. A dimensionless value $\sigma$ is used as the vertical coordinate, which is
specified as $\sigma = (z - \zeta) / (H - \zeta)$, where $z$ is the vertical coordinate; $\zeta = \zeta(\lambda, \varphi, t)$ – is the deviation of
the sea surface height (SSH) from the undisturbed surface as a function of longitude $\lambda$, latitude $\phi$, and time $t$; and
$H = H(\lambda, \varphi)$ is the sea depth (Diansky, 2013). The prognostic variables of the model are the horizontal
components of the velocity vector, potential temperature $T$, salinity $S$, and deviation of sea surface height from
undisturbed surface. The equation of state specially designed for the numerical models is used to calculate the
water density (Brydon, 1999).
The INMOM includes a sea ice module that takes into account the dynamics of the sea ice, ice melting, and
formation of sea ice and snow, as well as the transformation of old snow to sea ice (Yakovlev, 2003). This
module calculates the ice drift velocity, which depends on wind, sea currents, Earth's rotation, sea surface slope,
and ice floe interactions described by elastic-viscous-plastic rheology (Briegleb et al., 2004). The ice module
uses a monotonic transfer scheme (Hunke and Dukowicz, 1997), ensuring non-negative values of ice/snow
concentrations and mass. Detailed description of the basic configuration of the INMOM model can be found in
(Moshonkin et al., 2018).
The INMOM has been widely used in studies of the Black and Azov Seas (Zalesny et al., 2012; Fomin and
Diansky, 2018; Korshenko et al., 2019), the Norwegian Sea (Morozov et al., 2019), the Barents Sea (Diansky et
al., 2019), and the Sea of Okhotsk (Diansky et al., 2020). For this study, the INMOM model was run for two
years (2009–2010) within the region bounds of 9.4°E–30.4°E and 53.6°N–65.9°N using a uniform 2-mi grid in
the horizontal direction, non-uniform 35 sigma-levels in the vertical direction, and a 2.5-min calculation step.
The model outputs represent the 6-h averaged sea level height.
The Baltic Sea bottom topography was downloaded from the Baltic Nest Institute portal (http://nest.su.se). The
initial bathymetric product of 1' × 1' resolution was recalculated to match the 2-mi resolution of the model grid.
On the solid boundaries, no-normal flow and free-slip boundary conditions for momentum were applied, and the
heat and salt fluxes were set to zero.
The mean monthly water temperature and salinity fields provided by the Copernicus Marine Service Information
portal (http://marine.copernicus.eu) were used for model initialisation. This product represents the output of the
three-dimensional baroclinic hydrodynamic ocean model Hiromb-BOOS-Model (HBM-V1), assimilating the in
situ vessel and satellite observations. The data cover the 1990–2009 period and contain the sea level, current
velocity, temperature, and salinity with a 5.6-km horizontal and 5-m vertical resolution.
The INMOM model was forced using Era-Interim atmospheric reanalysis (Berrisford et al., 2011). The
reanalysis has a 0.75° spatial resolution and 6 h temporal resolution. The INMOM model used the following
forcing parameters: air temperature and humidity at an altitude of 2 m, atmospheric pressure at sea level, wind
speed of 10 m, precipitation, and short-wave and long-wave radiation.





The liquid boundary was drawn in the Kattegat Strait along 57.73°N (Fig 1) and defined using the Copernicus mean monthly values of sea temperature and salinity, as well as the hourly sea level records on two tide gauge stations, Frederikshavn (57.43°N, 10.57°E) and Goteborg Torshamnen (57.68°N, 11.79°E), located on the east and west coast of the strait, respectively. In situ sea level measurements from these stations were interpolated to the model grid nodes along the liquid boundary line.

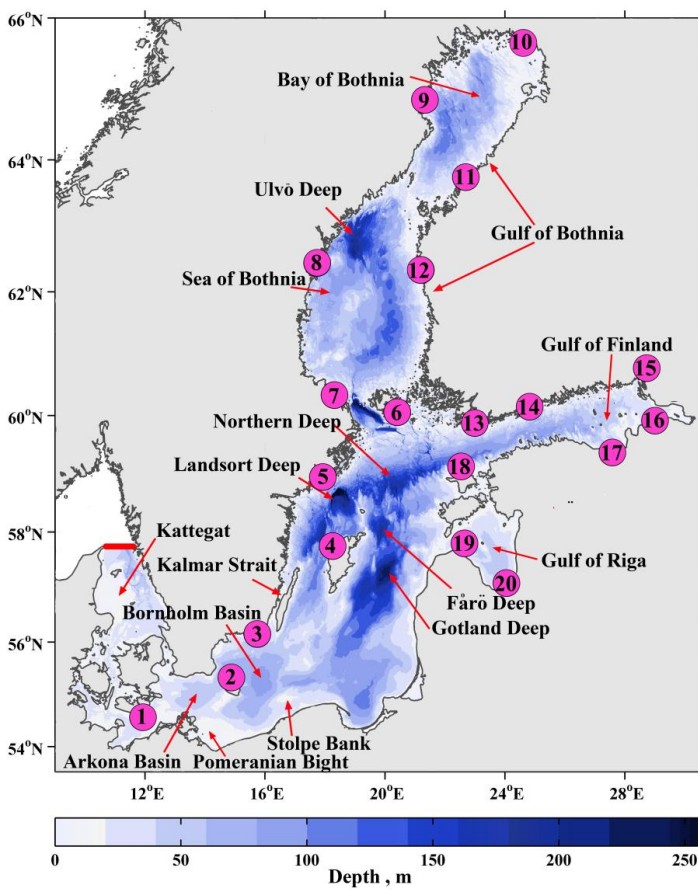

**Figure. 1. Bathymetry and location of tide gauge stations used for model validation. The liquid boundary of the modelled area in the Kattegat Strait is indicated by bold red line. The map is created using Baltic Sea Bathymetry Database (BSBD) http://data.bshc.pro/**

Water level observations at 20 other gauging stations (Fig. 1) served to validate the model outputs. In situ data were provided by the Copernicus Marine Service and the Northwest Hydrometeorological Service of Russia. Table 1 presents the metadata of the stations used for validation and includes the station coordinates, sea level measurement frequency, number of sea level measurements used in this study, and percentage of missing data. The in situ time series are sufficient for the validation exercise and have only a few gaps. The percentage of missing data (found only for six stations) does not exceed 6.1%. For the validation procedure, the in situ observations were averaged to match the 6 h output frequency of the model.





**Table 1. Tide gauge stations used in the study.**

| No | Station name | Period | Coordinates | | Measuring span | Number of measurements | Missing values,% |
|----|-------------|--------|-------------|-----|----------------|------------------------|------------------|
| | | | Lat. (°N) | Lon. (°E) | | | |
| 1 | Gedser | 2009- 2010 | 54.57 | 11.93 | 10 min | 104093 | 1.3 |
| 2 | Tejn | 2009-2010 | 55.25 | 14.83 | 1 h | 17338 | 1.0 |
| 3 | Kungsholmsfort | 2009-2010 | 56.11 | 15.59 | 1 h | 17520 | 0.0 |
| 4 | Visby | 2009- 2010 | 57.64 | 18.28 | 1 h | 17520 | 0.0 |
| 5 | LandsortNorra | 2009- 2010 | 58.77 | 17.86 | 1 h | 17520 | 0.0 |
| 6 | Degerby | 2009-2010 | 60.30 | 20.38 | 1 h | 17520 | 0.0 |
| 7 | Forsmark | 2009- 2010 | 60.41 | 18.21 | 1 h | 17520 | 0.0 |
| 8 | Spikarna | 2009-2010 | 62.36 | 17.53 | 1 h | 17520 | 0.0 |
| 9 | Furuogrund | 2009- 2010 | 64.92 | 21.23 | 1 h | 17520 | 0.0 |
| 10 | Kemi | 2009-2010 | 65.67 | 24.52 | 1 h | 17520 | 0.0 |
| 11 | Pietarsaari | 2009- 2010 | 63.71 | 22.69 | 1 h | 17520 | 0.0 |
| 12 | Kaskinen | 2009-2010 | 62.34 | 21.21 | 1 h | 17520 | 0.0 |
| 13 | Hanko | 2009- 2010 | 59.82 | 22.98 | 1 h | 17520 | 0.0 |
| 14 | Helsinki | 2009-2010 | 60.15 | 24.96 | 1 h | 17520 | 0.0 |
| 15 | Vyborg | 2009- 2010 | 60.70 | 28.73 | 1 h | 17520 | 0.0 |
| 16 | Schepelevo | 2009- 2010 | 59.99 | 29.15 | 1 h | 17520 | 0.0 |
| 17 | Sillamae | 2009-2010 | 59.42 | 27.74 | 1 h | 16810 | 4.1 |
| 18 | Lehtma | 2009- 2010 | 59.07 | 22.70 | 1 h | 16465 | 6.1 |
| 19 | Kolka | 2009-2010 | 57.73 | 22.58 | 1 h | 16520 | 5.7 |
| 20 | Daugavgriva | 2009-2010 | 57.05 | 24.02 | 1 h | 17102 | 2.4 |


**2.2 Model validation**

The sea level simulated by the basic INMOM configuration was verified against the in situ observations using a
set of standard statistics: absolute ($\sigma_{abs}$) and relative ($\sigma_{rel}$) bias, root mean square error ($\sigma_{er}$), and correlation
coefficient (R). The standard deviation of the observed ($\sigma_m$) and simulated ($\sigma_{tg}$) SSH, as well as their relation
($\sigma_p$), were evaluated, and the additional criteria of accuracy ($P_m$, %) were introduced. These criteria allow the
assessment of the number of good simulations considering the accuracy $< 0.674\sigma_{tg}$.
$$\sigma_{abs} = \frac{\sum_{i=1}^{N}\left|\zeta_m - \zeta_{tg}\right|}{N}$$

148                                                                                              (1)

where $N$ is the time series length, $\zeta_m$ is the modelled sea level, and $\zeta_{tg}$ is the tide gauge observations.
$$\sigma_{rel} = \frac{\sigma_{abs} * 100\%}{(\zeta_{tg})_{max} - (\zeta_{tg})_{min}}$$

                                                                                              (2)

where $(\zeta_{tg})_{max}$ is the maximum and $(\zeta_{tg})_{min}$ is the minimum value of the in situ observations.




$$\sigma_{er} = \sqrt{\frac{\sum\limits_{i=1}^{N}(\zeta_m - \zeta_{tg})^2}{N-1}}$$

(3)

$$\sigma_m = \sqrt{\frac{\sum\limits_{i=1}^{N}(\zeta_m - \overline{\zeta_m})^2}{N-1}}$$

(4)

$$\sigma_{tg} = \sqrt{\frac{\sum\limits_{i=1}^{N}(\zeta_{tg} - \overline{\zeta_{tg}})^2}{N-1}}$$

(5)

where $\overline{\zeta_m}$ is the mean modelled and $\overline{\zeta_{tg}}$ is mean observed sea level.

$$\sigma_p = \frac{\sigma_{er} *100\%}{\sigma_{tg}}$$

(6)

$$R = \frac{\dfrac{1}{N-1}\sum\limits_{i=1}^{N}(\zeta_{tg} - \overline{\zeta_{tg}})(\zeta_m - \overline{\zeta_m})}{\sigma_{tg}\sigma_m}$$

(7)

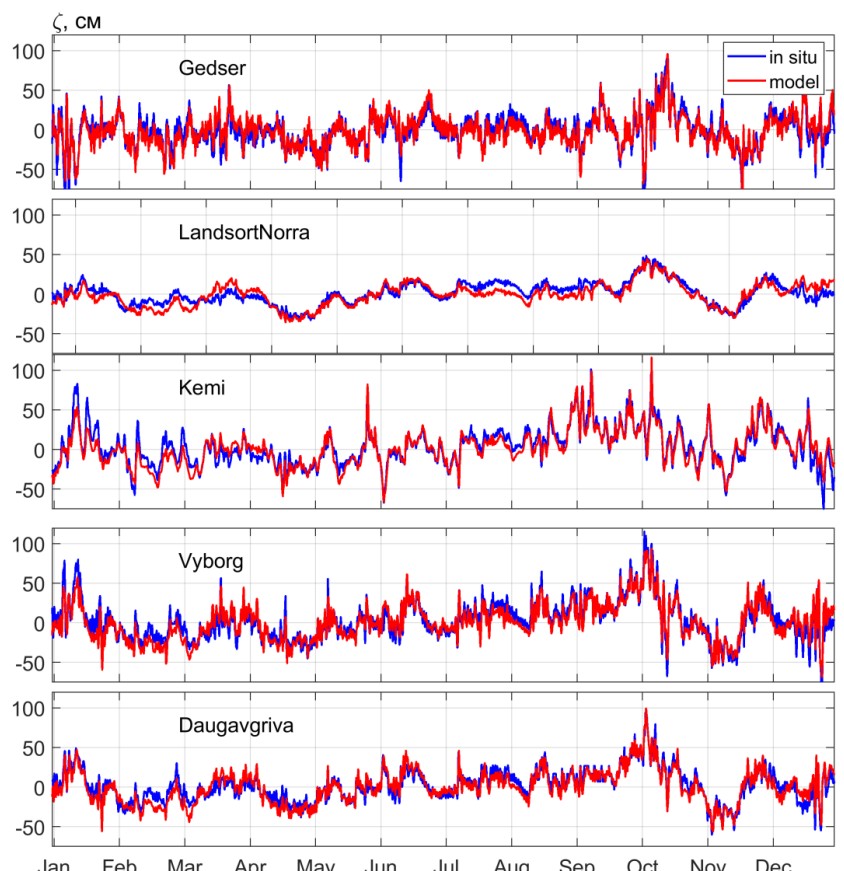

**Figure 2. Time series of in situ (blue) and modelled (red) sea level for 2009. The modelled dataset is derived from the basic configuration of the INMOM model (see Section 2.1).**






A comparison of the SSH model outputs and the observations from the gauging stations (Fig. 2) demonstrates
that the model reproduces the sea level variations in different parts of the Baltic Sea well. The correlation
between the simulated and observed time series was higher than 0.79. The absolute bias ranges within 6.7–9.2
cm, which represents 3.7–7.4% of the SSH magnitude at the gauging stations. Most of the model outputs (from
75% to 90%) have considerably good accuracy (Pm < 0.674σtg).

**2.3 Modelling free sea level oscillations in barotropic and baroclinic conditions**

To investigate the difference between free sea level oscillations in barotropic and baroclinic conditions, the
INMOM model was run again in two different configurations.
In the barotropic configuration, the salt and heat fluxes were set to zero and the water density in the sea state
equation depended only on pressure. In the baroclinic configuration, the INMOM model took into account both
salt and heat fluxes, and the water density varied with pressure, temperature, and salinity. In both the barotropic
and baroclinic implementations, the Baltic Sea was considered a fully enclosed basin, with no water exchanged
with the North Sea. The liquid border in the Kattegat Strait was assumed to be solid. River water input and ice
conditions were also neglected.
Under natural conditions, the free sea level oscillations attenuate rapidly due to the dissipative effects of vertical
and horizontal viscosity, near-bottom friction, non-linear effects, and Earth's rotation (Proshutinsky 1993,
Zakharchuk et al., 2004). According to previous numerical experiments (Proudman, 1953; Wübber and Kraus,
1979; Zakharchuk et al., 2004), the relaxation of the Baltic large-scale free sea level oscillations takes several
days. Setting the turbulent viscosity to zero for the vertical components and to the minimum values for the
horizontal components allows the damping of the simulated sea level fluctuations to be reduced. Because of this
modification, a spectrum of sea level oscillations at lower frequencies can be estimated with significantly higher
resolution.
In both the barotropic and baroclinic numerical experiments, the model was perturbed for 10 days (1–10 January
2009) using Era-Interim reanalysis. The meteorological forcing was then turned off and the simulations were run
for 2 years (2009–2010) considering only free dynamic oscillations.


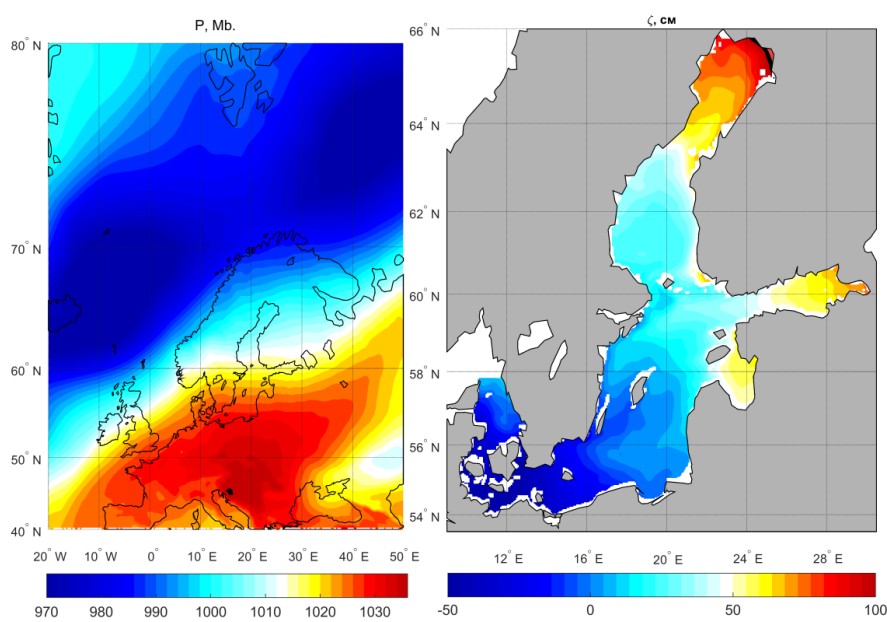


**Figure 3. ERA-Interim atmospheric pressure (a) and INMOM SSH (b) for the moment of cessation of atmospheric**
**forcing (10 January 2009, 18 h).**

At the end of the atmospheric forcing and the beginning of free sea level simulations, the southern part of the
Baltic Sea was under an atmospheric anticyclone centred over Central Europe, while the northern part of the sea
was affected by a low-pressure system that had developed over the Norwegian Sea. These meteorological
conditions resulted in prevailing western winds (Fig. 3a), and finally led to a 50–100 cm sea level increase in the
north and east and to 30–50 cm sea level decrease in the southwest (Fig. 3b) parts of the Baltic Sea.

Fourier analyses of the simulated SSH time series were performed using the following decomposition.

$$f(t) = Z_0 + \sum_{k=1}^{N/2} (a_k \cos k\omega t + b_k \sin k\omega t), \qquad \left( \omega = \frac{2\pi}{T}, k = 0, 1, 2, ... \right) \tag{8}$$

where f(t) is the sea level time series, N is the time series length, T is the period, t is the time, $a_k$ is the
coefficient at frequency $\omega$, $Z_0$ is the mean average of the sea level time series, and k is the coefficient number.

The phase ($F_k$) and amplitude ($A_k$) were calculated using Equation (9) for each model node, and their spatio-
temporal distribution was analysed.

$$A_k = \sqrt{a_k^2 + b_k^2}, \quad F_k = \arctan\left(b_k/a_k\right) \tag{9}$$

The wave phase velocity (C) was estimated using the phase difference between adjacent nodes:

$$C_x = \frac{\Delta x}{P \Delta F_x}, \quad C_y = \frac{\Delta y}{P \Delta F_y}, \tag{10}$$

$$C = \sqrt{C_x^2 + C_y^2} \tag{11}$$




where Cx and Cy are zonal and meridional components of the wave phase velocity, ΔFx and ΔFy are the zonal
and meridional phase difference, respectively, and P is the period.
The estimation of the phase speed was performed only for regions where Ak > 0,67σ (Guide to Marine
Hydrological Forecasts, 1994).
$$\sigma = \sqrt{\frac{A^2}{2}}$$    (12)
where $A$ is the field average sea level amplitude at each frequency $\omega$.

**3 Comparison of simulations of free barotropic and baroclinic sea level oscillations**

**3.1 Free barotropic oscillations**

In general, simulated free barotropic sea level oscillations in the Baltic Sea are low and range within 3–15 cm
depending on the region (Fig. 4). The maximum amplitudes are noted in the eastern Gulf of Finland. The
minimum values occur principally in the central part of the Baltic Proper.

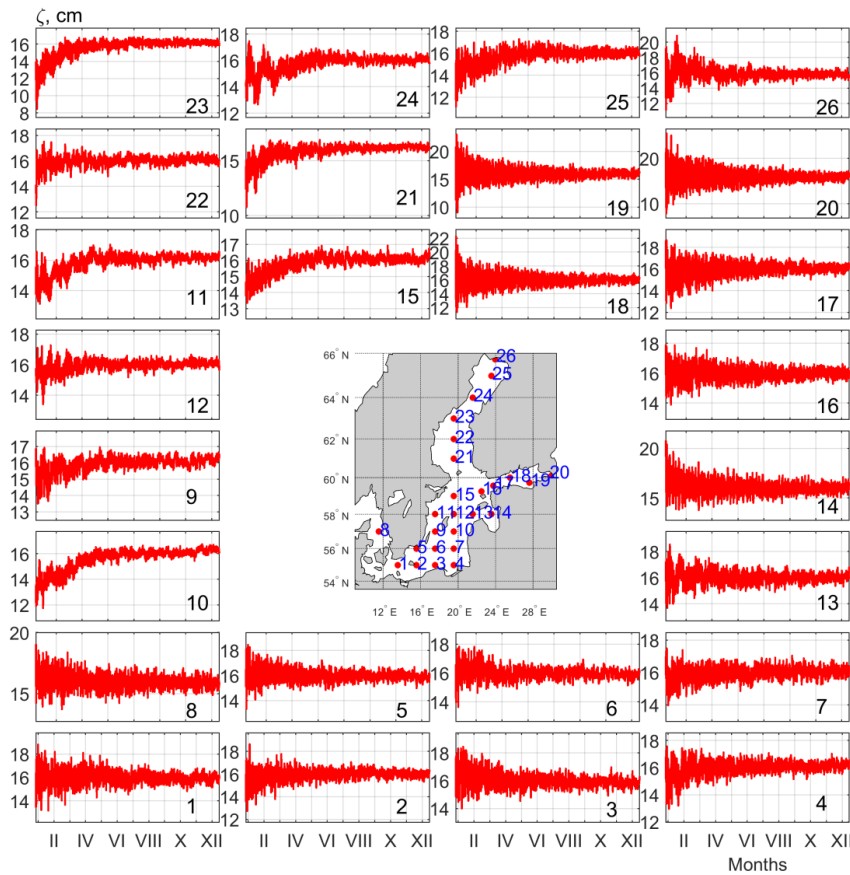

**Figure 4. Time series of free barotropic sea level oscillations at selected points simulated by INMOM model. Location**
**of points is shown on the map.**





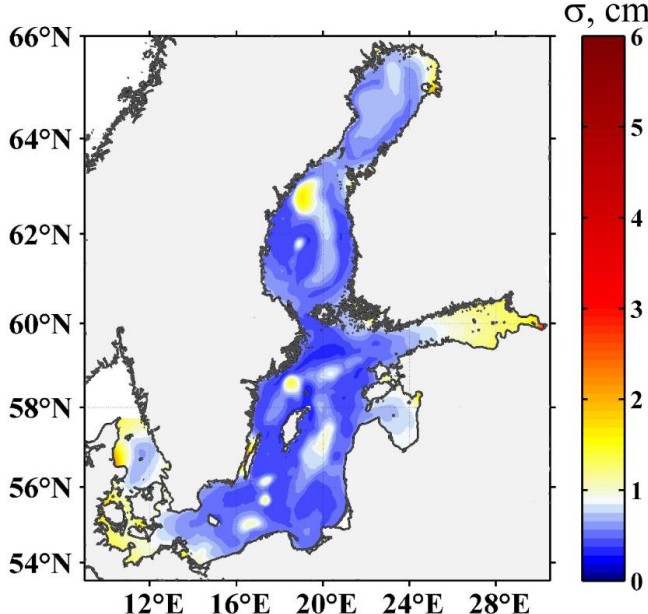

**Figure 5. Standard deviation (σ) of amplitudes of free barotropic sea level oscillations simulated by INMOM model.**

The standard deviation (σm) of the sea level amplitudes estimated for each grid node can be used for the characterisation of the oscillation intensity. The spatial distribution of the σm values demonstrates that the highest barotropic oscillations (σm of 2.5–5 cm) can be found in the Neva Bay of the Gulf of Finland, in the northern Bay of Bothnia near Hailuoto Island, as well as in the Kalmar Strait near the southeast Sweden coast (Fig. 5). Barotropic oscillations of medium amplitude ($\sigma_m$ of 1–2 cm) are observed in the Pärnu Bay of the Gulf of Riga, northeast of the Baltic Proper, near Rügen Island, as well as in the Danish straits and Kattegat Strait. Oscillations of medium intensity can be noted as over local uplifts in the Baltic Proper as over-bottom depressions, such as the Ulvö Deep in the Bothnia Sea and Landsort, Northern, and Gotland Deeps. These local spots have not been observed in previous experiments to be effectuated using a shallow-water model (Jönsson et al., 2008). In the shallow-water equations, the water movement is independent of the vertical coordinate. Sea level fluctuations are generated only due to full flux divergence and surface slope related to geostrophic balance. In regions with sharp bathymetry (uplifts, sills, deeps), the generation of relatively high perturbations of the vertical component of the speed of barotropic flux is probable. These perturbations may not be negligible and, presumably, affect sea level fluctuations.

The Fourier analysis of the simulated time series of free barotropic sea level oscillations (Fig. 6) indicates that amplitude peaks frequently occur at periods of 13, 15–16, 19, 23, 27, 29, 41, and 44 h. Near the Gulf of Finland and in the southeast Baltic Proper, the period of the highest amplitude peak is 13 h. In the inner Gulf of Finland, oscillations of 27-h periods became prevalent. The barotropic free oscillations of this period dominate in the northern Gulf of Bothnia and south-western Baltic Proper. Other significant oscillations of 15, 23, 29, and 41 h are also observed in the Gulf of Finland. However, their amplitude is 2–4 times lower than that of 27 h period oscillations.





In the south-eastern and eastern Baltic Proper, free barotropic oscillations of 13 and 41 h periods have the
highest amplitudes. In the centre of the Bothnia Sea, the dominant oscillation has a 19.5-h period. In the Gulf of
Riga, the highest amplitude was observed as 23 h barotropic oscillations. This result differs from the 17-h period
found in the study by Jönsson et al. (2008) based on the shallow-water model.

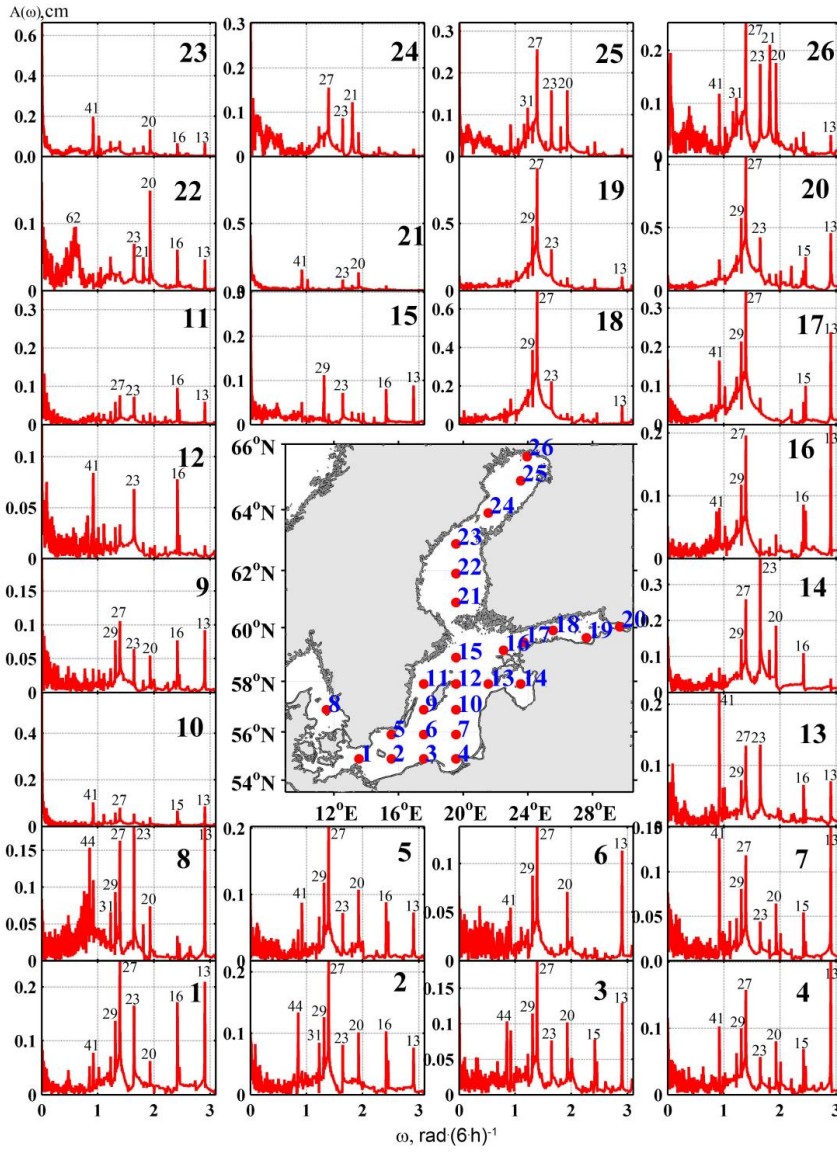


**Figure 6. Amplitude spectra A(ω) of free barotropic oscillations in different parts of the Baltic Sea. Numbers above**
**the peaks show oscillation periods in hours. Locations of points are shown on the map.**

Nevertheless, a portion of our results is consistent with those determined by a numerical experiment conducted
by Wübber and Krauss (1979), where, similar to our study, the effect of the Earth's rotation was taken into





account. These authors identified eigenoscillations with periods of 31.0, 26.4, 22.4, 19.8, 17.1, and 13.0 hours. In
our experiment, the corresponding periods were 31, 27, 23, 20, 17, and 13 hours. Moreover, owing to the more
sophisticated 3-D model, higher spatial resolution of the grid, and longer period of simulations (716 days), we
were able to improve both the spectral resolution of the simulated time series and their spectral range. We
identified additional free barotropic oscillations of periods of 44, 41, 37, 29, 21, 16, and 15 h that have not been
noted previously.

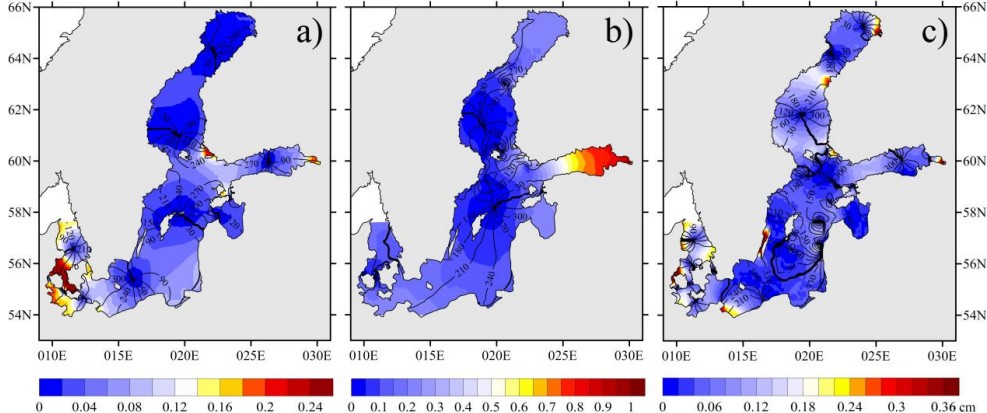


**Figure 7. Maps of amplitudes (in cm) and phases in degree (isolines) of free barotropic sea level oscillations with 13 h**
**(a), 27 h (b), and 41 h (c) periods.**

An analysis of the spatial distribution of the amplitude and phase of free barotropic oscillations with periods of
13, 27, and 41 h (Fig. 7) demonstrates that due to the Earth's rotation and the enclosed configuration of the sea,
these oscillations transform into progressive-standing waves (PSW). Similar to the amphidromic systems of tidal
waves (Nekrasov, 1975; Pugh, 1987; Voynov, 2003), there are no sea level oscillations in the PSW nodes, while
in the PSW, the oscillations are maximised. The progressive-standing waves of 13 h periods have 10 nodes (Fig.
7a). Their maximum amplitudes are observed in their antinodes located in the Danish straits and eastern Gulf of
Finland.
The location of our 13 h amphidromic systems in the Gulfs of Bothnia, Finland, and Riga agrees well with the
results found by Wübber and Krauss (1979) for the 13.04 h eigenoscillations. However, for the Baltic Proper,
our systems (near the Fårö and Bornholm Deeps) are shifted by 200 km toward the northeast. Another significant
difference is the direction of isophase rotation, which in our experiment occurs clockwise, while Wübber and
Krauss suggested an anticlockwise rotation.
Free barotropic oscillations of 27 h periods have two predominant amphidromic systems: one is in the Bothnia
Sea and the second is to the northeast of Gotland Island (Fig. 7b). Their location is consistent with the location of
the corresponding eigenoscillation of the 26.4 h period of Wübber and Krauss. Our simulations allowed the
detection of several more degenerate amphidromic systems of 27 h periods, which have not been reported by
previous studies. Degenerate amphidromic systems were found in the northern Bothnia Sea, the Aland Sea, the
central and south-eastern parts of the Gulf of Bothnia, at the exit of the Gulf of Finland, and in the Great Belt and
Sound Straits. The PSW antinodes with 27 h periods have variable amplitudes, with the highest amplitude
located in the eastern Gulf of Finland. Antinodes with lower amplitudes are situated in the northern Gulf of



Bothnia, to the southeast of the Aland Islands, in Pärnu Bay of the Gulf of Riga, and in the south-western Baltic
Proper. In contrast to the 13 h amphidromic systems, the isophase rotation in the main 27 h systems occurs
anticlockwise.
Free barotropic oscillations of 41 h periods are characterised by larger amounts of amphidromic systems (Fig.
7c). Their primary systems are detected in the northern and central Gulf of Bothnia, Bothnia Sea, eastern Gulf of
Finland, Kattegat Strait, and Danish straits. Numerous degenerate amphidromic systems can be seen in the north,
east, and central parts of the Baltic Proper, along its southern coast, the easternmost part of the Gulf of Finland,
and in the Danish Straits. The amplitude of the free 41 h oscillations is 2 times lower than that of the 27 h period
waves. The most noticeable PSW antinodes are localised within the narrow areas of the coastal zones in the
northern and eastern parts of the Gulf of Bothnia, in the Neva Bay of the Gulf of Finland, along the western,
eastern, and southwestern coasts of the Baltic Proper, as well as in the Danish and Kattegat Straits. The isophases
of 41 h oscillations rotate in a clockwise direction, similar to the 13 h period waves.

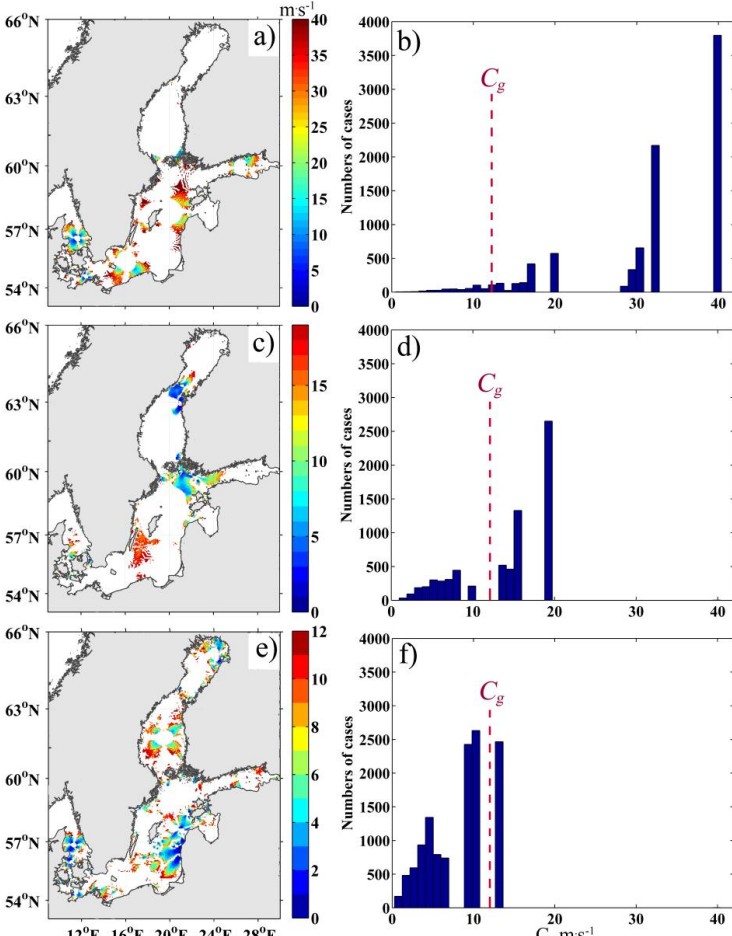


**Figure 8. Maps and histograms of phase speed of progressive-standing waves of 13 h period (a,b), 27 h period (c,d)**
**and 41 h period (e,f). Dash line on the histogram plots indicates minimum theoretical value of phase speed of**
**barotropic gravity wave (Cg).**



An estimation of the phase speed ($C$) of the PSW using Equations (10) and (11) demonstrates that $C$ reduces
with an increase in the wave period (Fig. 8). For the 13 h PSW, the phase speed can reach 40 m s$^{-1}$, for the 27 h
PSW, only 19 m s$^{-1}$, and for the 41 h PSW, it can reach 13 m s$^{-1}$.
The average depth of the Baltic Sea and its main gulfs varies from 23 to 77 m, while the maximum values reach
51–459 m (Lepparana and Myrberg, 200). Under these conditions, the theoretical phase speed of the barotropic
gravity wave in the Baltic Sea, calculated using Equation (13), ranges between 12 and 67 m s$^{-1}$.
$$C_g = \sqrt{gH} ,$$ (13)
where $H$ is the depth and $g$ is the acceleration due to gravity.
Most of our $C$ estimates for the 13 h waves are within this theoretical range (Fig. 8b). For only 70% of the
detected 27 h waves, the phase speed agrees with the theoretical values (Fig. 8d), while among the PSWs of 41 h
period, waves that are lower than the theoretical phase speed dominate (Fig. 8f).

**3.2 Free sea level oscillations in baroclinic conditions**

In stratified basins along with high-frequency (daily and hourly scales) oscillations, the low-frequency free
oscillations of the seasonal scale are also generated after the anemobaric forcing ceases. These observations have
periods from several months to one year and reach 30–35 cm in amplitude (Fig. 9).

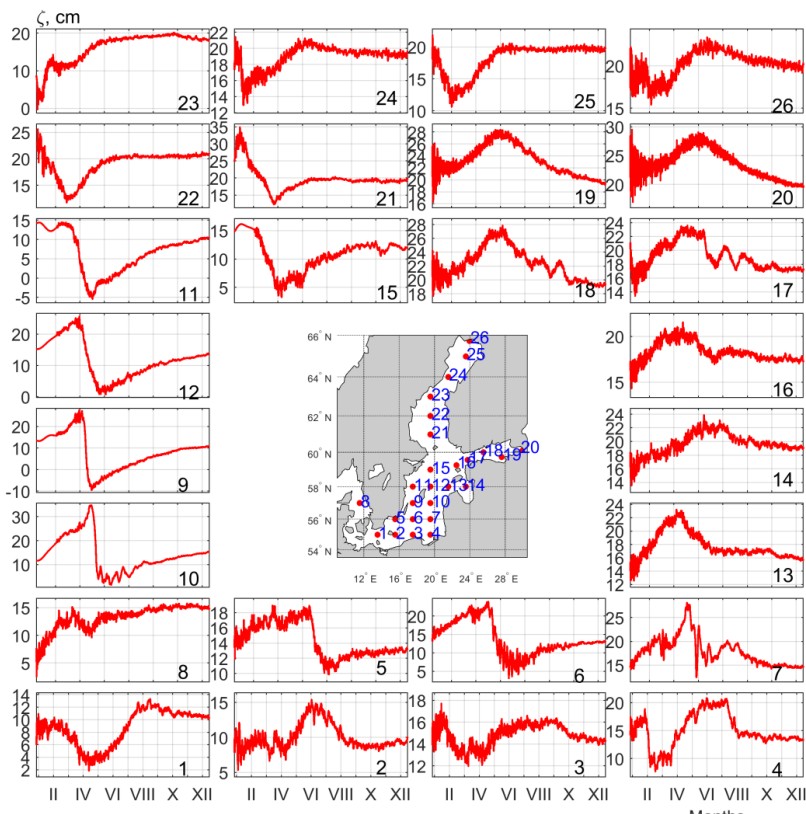






**Figure 9. Time series of free baroclinic sea level oscillations at selected points simulated by INMOM model. Location**
**of points is shown on the map.**

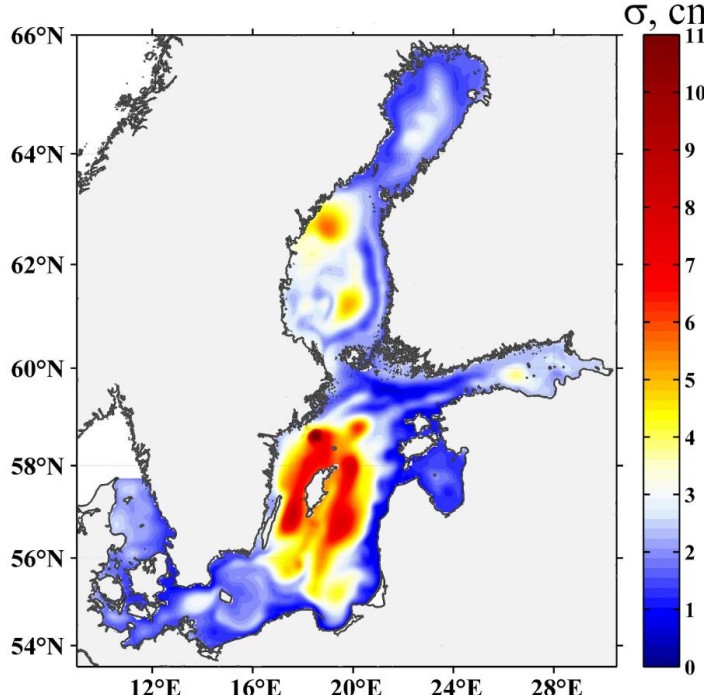


**Figure 10. Standard deviation (σ) of amplitudes of free sea level oscillations in the baroclinic sea, simulated using**
**INMOM model.**

The spatial distribution of the standard deviation of the amplitudes of the free oscillations in the baroclinic sea
(Fig. 10) demonstrates that the location of the zones with a high SSH variability is similar to that found in the
barotropic experiment. These are the deep-water basins of the Baltic Proper: Landsort Deep, Farö Deep,
Northern Deep, and Gotland Deep, as well as the Ulvö Deep in the Sea of Bothnia. The $\sigma_m$ values in the
baroclinic experiment were 4–6 times higher than in the barotropic study. We also identified several zones of
moderate SSH variability, which were not detected in the barotropic simulations. They are situated in the south-
eastern Sea of Bothnia, central Gulf of Finland (off the Narva Bay), the straits between the Ellesmere and
Gotland Islands, and central Arkona Basin.
The Fourier analysis demonstrates that in baroclinic conditions, the maximum energy concentrates mostly at low
frequencies. However, the differentiation of distinct peaks in low-frequency bands is problematic (Fig. 11).

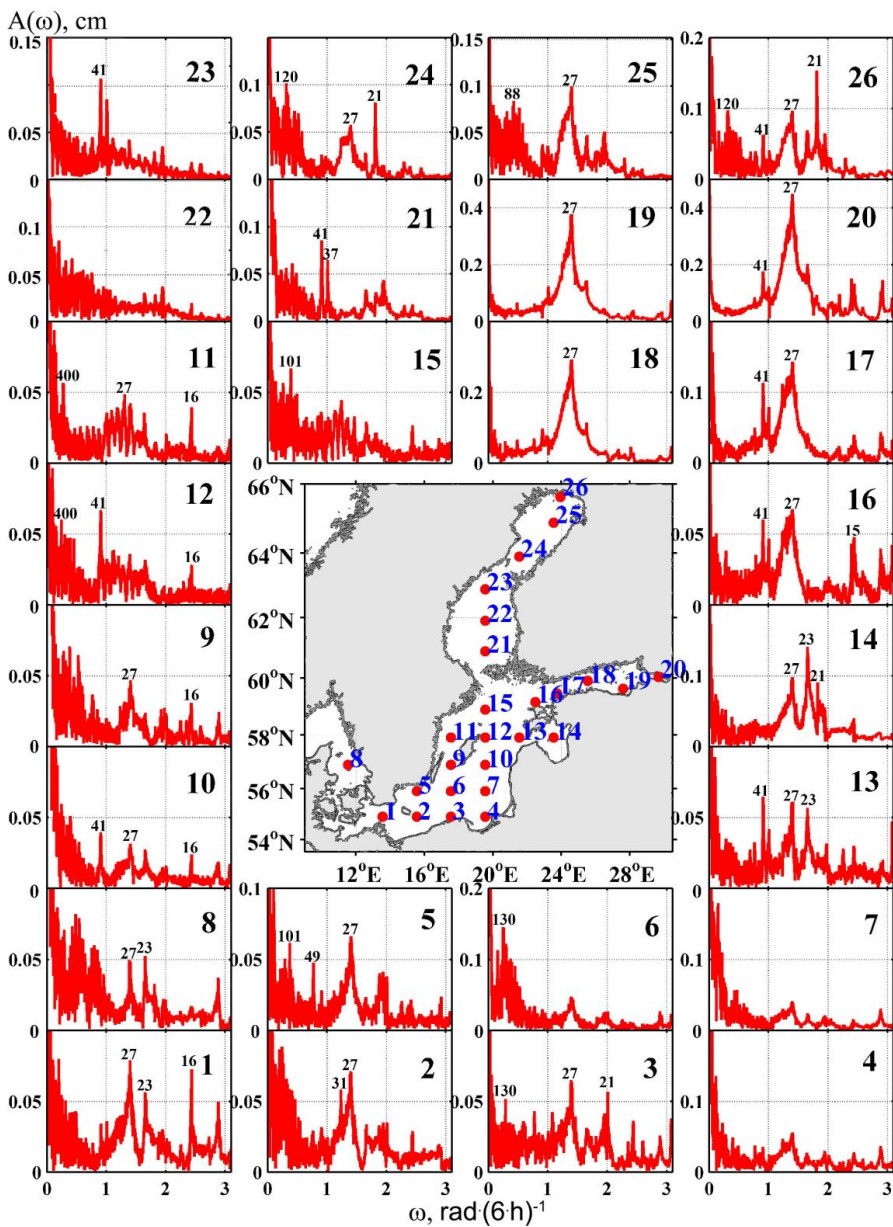

**Figure 11. Amplitude spectra A(ω) of free oscillations in baroclinic conditions in different parts of the Baltic Sea. Numbers above the peaks show oscillation periods in hours. Locations of points are shown on the map.**

In higher frequencies, the energetic maximums correspond to those found in the barotropic experiments (e.g. to peaks with periods of 13, 19, 23, 27, and 41 h). The difference with the barotropic experiment consists of a decrease in the peak amplitude along with an increase in width. This difference could be explained by the following two factors: 1) the stratification for barotropic free sea level oscillations can work as a dissipative


factor; 2) when a barotropic current interacts with sharp bathymetry, the vertical component of the current
significantly increases. This component affects a pycnocline and generates baroclinic oscillations with
frequencies close to the barotropic results. The resulting oscillations became amplitude-modulated and their
spectral peaks broadened.

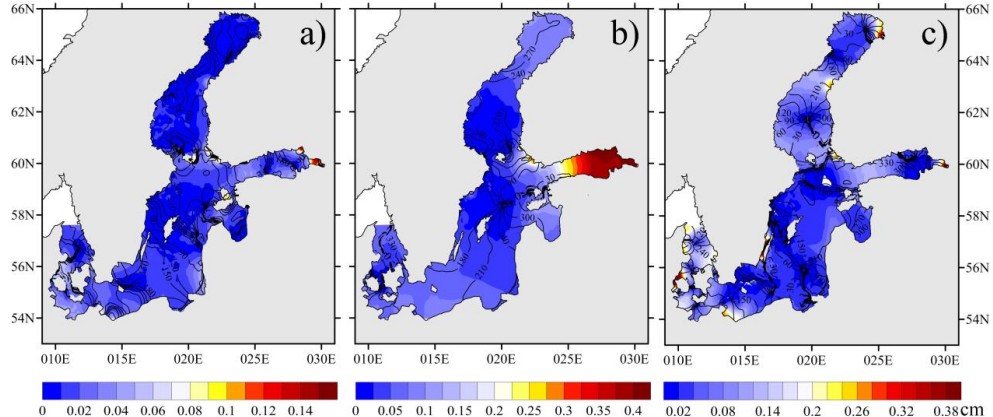


**Figure 12. Maps of amplitudes (in cm) and phases in degrees (isolines) of free sea level oscillations with 13 h (a), 27 h**
**(b) and 41 h (c) periods in the baroclinic sea.**

For comparison with the barotropic experiment, the spatial distribution of the amplitudes and phases of the 13 h,
27 h, and 41 h oscillations in baroclinic conditions is shown in Fig. 12. In a stratified sea, the amplitude of the 13
h and 27 h oscillations is two times lower. For lower-frequency waves (41 h), the difference with barotropic
conditions is negligible. The highest amplitudes for the 13 h periods are observed in the eastern Gulf of Finland
and in Vyborg Bay (Fig. 12a). In the stratified environment, the 13 h amphidromic systems disappear in the Gulf
of Bothnia and Gulf of Finland, as well as in the central Baltic Proper. The systems remain detectable only in the
southern Baltic Sea and in the Kattegat Strait. Free oscillations of 27 h periods in the baroclinic conditions
reached the maximum in the narrow zone near the southwest Finland coast (Fig. 12 b). The 27 h amphidromic
system is observed only in the central part of the Baltic.
The spatial structure of the 41 h free oscillations in the baroclinic conditions was similar to that found in the
barotropic experiment. The oscillations of higher intensity are observed within small coastal areas in the north
and east of the Gulf of Bothnia, in the Neva Bay of the Gulf of Finland, along the west and southwest coasts of
the Baltic Proper, as well as in the Danish and Kattegat Straits (Fig. 12 c). The location of the 41 h amphidromic
systems in the baroclinic conditions in many areas (north of the Gulf of Bothnia, the Bothnia Sea, east of the
Gulf of Finland, the Kattegat Strait, and the Danish straits) is similar to that found in the barotropic experiment.
However, in stratified conditions, the degenerate amphidromic systems change. One system in the east of the
Baltic Proper disappears, while a new appears in the south-eastern section of the sea (Fig. 12 c).
The phase speed of the PSW movement in the baroclinic conditions varies within 2–37 m s$^{-1}$ for the 13 h waves,
1–20 m s$^{-1}$ for the 27 h waves, and within 1–13 m s$^{-1}$ for the 41 h waves (Fig. 13).
To interpret the detected free-sea level oscillations in baroclinic conditions, we compared the estimated phase
speed of the modelled oscillations with the theoretical phase speed values of the baroclinic gravity waves. The
theoretical dispersion relation of an internal gravity wave ($C_i$) calculated for the 1.5-layer model (Carmack and



Kulikov, 1998) can be estimated using Equation (13), where $g$ is replaced by $g' = \dfrac{\Delta\rho}{\rho} g$ ($\rho$ is mean sea water
density, $\Delta\rho$ is difference in the densities between two layers, and $h'$ is upper sea layer depth).

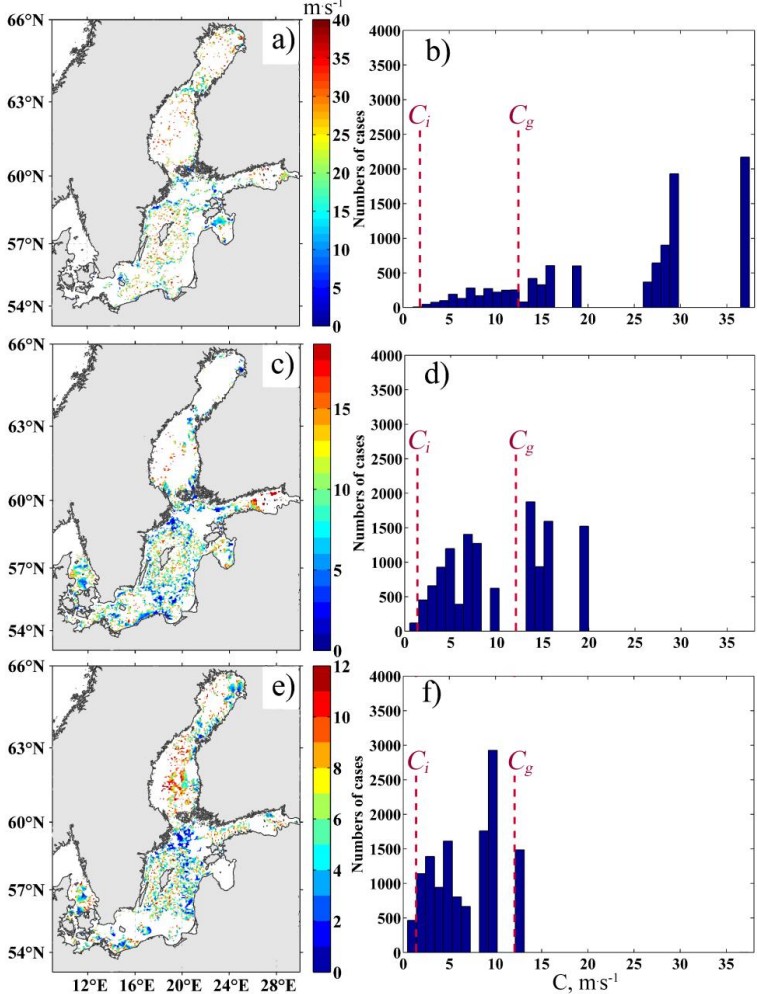

**Figure 13. Maps and histograms of phase speed of progressive-standing waves of 13 h period (a,b), 27 h period (c,d)**
**and 41 h period (e,f) in the baroclinic sea. Dash line on the histogram plots indicates minimum theoretical value of**
**phase speed of baroclinic (Ci) and barotropic (Cg) gravity waves.**
Using the Copernicus data of the vertical distribution of sea water density for 2009–2010, we first evaluated the
variables of Equation (13) and then estimated the phase speed of the internal gravity waves ($C_i$) for the entire
Baltic Sea. For variables ranging from 2 to 60 m ($h'$), 1.5–62 × 10$^{-4}$ ($\Delta\rho/\rho$), and 2–61 × 10$^{-3}$ m s$^{-1}$ ($g'$), the phase
speed of the internal gravity waves must vary within 0.08–1.53 m s$^{-1}$.
Our estimations of the phase speed ($C$) of free oscillations in the baroclinic medium using Equations (10) and
(11) for waves with 13, 27, and 41 h periods do not coincide with the range of theoretical phase speeds of the




internal gravity waves ($C_i$) in the Baltic Sea. Most of our $C$ estimations for the 13 h, 27 h, and a significant
portion of the 41 h baroclinic oscillations are within the range of theoretical values calculated for the barotropic
conditions (see Section 3.1).

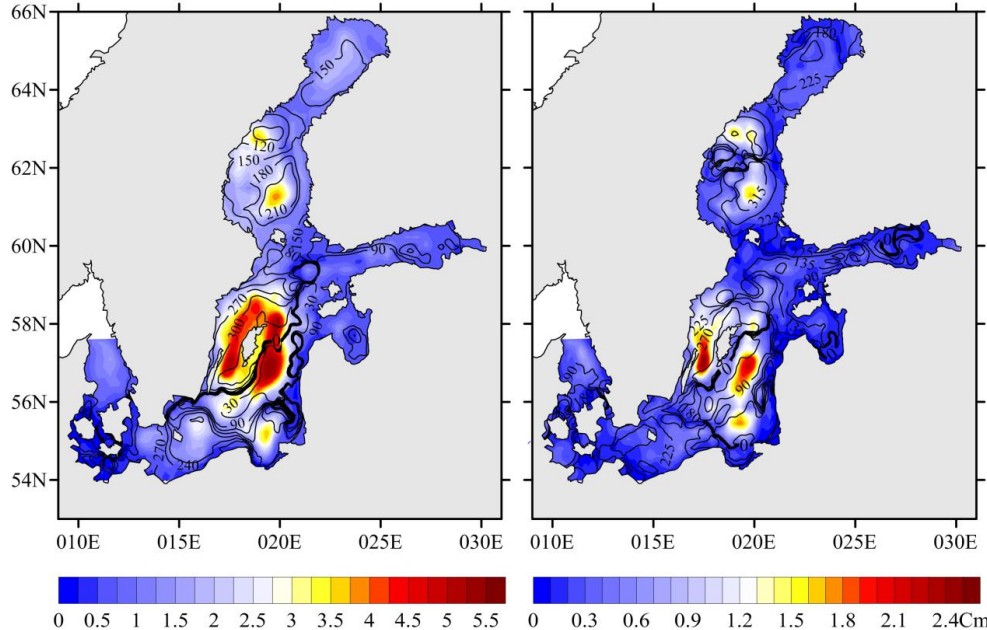


**Figure 14. Amplitudes in cm (colour) and phases in degrees (co-tidal lines) of the free sea level oscillations in the**
**baroclinic sea on periods 358 (a) and 89 (b) days.**

The spatial structure of free baroclinic oscillations of 89 days and 358 days (Fig. 14) agrees well with the spatial
distribution of the standard deviation of the amplitudes of the free oscillations in the baroclinic sea (Fig. 10).
This means that the overall spatial structure of the free oscillations in baroclinic seas is determined mostly by
oscillations at seasonal scales. The highest amplitudes of the long-period waves are observed in the deep regions
of the Baltic Proper and Bothnia Sea. Moreover, a significant spatial variability in their phases can be noted.
Nodal lines of these waves traverse the sea between the coasts in different parts. In areas of isophase
condensation, where the amplitudes of sea level oscillations are near 0, the phase can reverse to the opposite. In
other areas, the phase of 358 day oscillations can change gradually. This confirms the likely presence of a low-
frequency progressive component of wave movement, which is oriented mostly in the southern direction (Fig. 14
a).
Free baroclinic oscillations of 89 days have degenerate amphidromic systems in the southwest, south, and
northwest Baltic Proper. These systems rotate in a anticlockwise direction (Fig. 14 b). The phase velocity of the
seasonal PSWs vary within 0.01–0.07 m s$^{-1}$ and within 0.01–0.24 m s$^{-1}$, respectively for 358-day and 89-day
oscillations. Regarding the theoretical phase speed of the internal gravity waves ($C_i$), these values are
significantly lower for longer waves and belong to the theoretical range for waves of the shorter period(Fig.15).



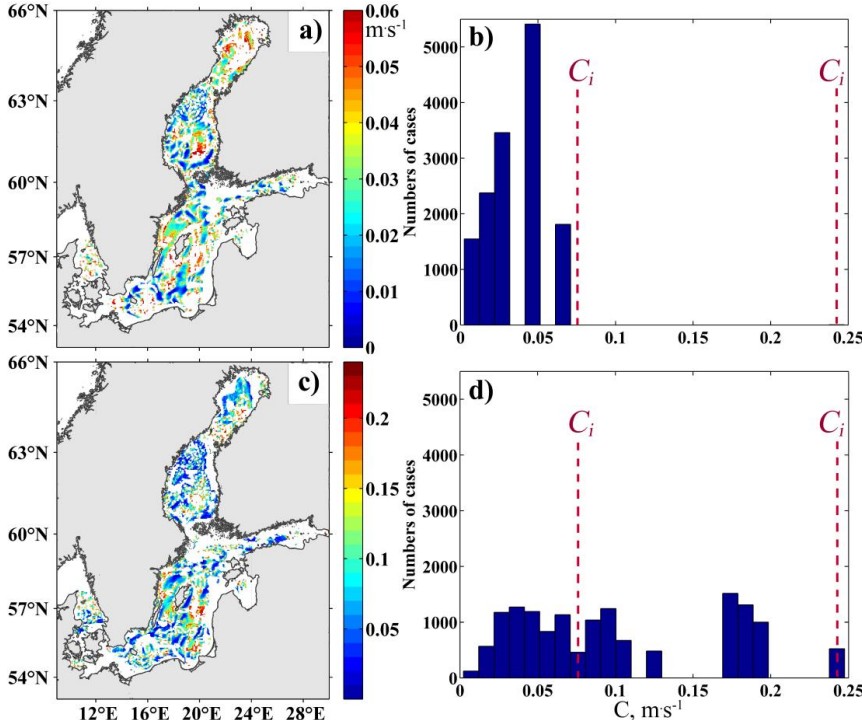

**Figure 15. Maps and histograms of phase speed (m s⁻¹) of progressive-standing waves of 385-day period (a,b) and 89-day period (c,d). Dash line on the histogram plots indicates minimum (left) and maximum (right) value of theoretical phase speed of baroclinic gravity waves estimated by (Eq. 13)**

## 4 Discussion

Our numerical experiments based on a three-dimensional hydrodynamic model demonstrated that after the cessation of anemobaric forces, the return of the Baltic Sea water mass to equilibrium in barotropic and baroclinic conditions is different.

In barotropic conditions, the maximum dispersion of free oscillations occurs on a time scale of 13, 15–16, 19, 23, 27, 29, 41, and 44 h. The highest oscillations with amplitudes of 7.5 cm occur in the head of the Gulf of Finland, Gulf of Bothnia, and in the Kalmar Strait.

In baroclinic conditions, high-frequency free oscillations (periods of 13, 19, 23, 27, and 41 h) are also observed. However, their role is minor, with amplitudes that are significantly lower than the amplitude of lower-frequency oscillations. In baroclinic conditions, oscillations of periods from several months to one year with amplitudes of 15–17 cm appear. The area with the highest amplitudes of free baroclinic oscillations moves to the deep part of the Baltic Proper, where the highest gradients of water density are observed (Hydrometeorology and Hydrochemistry of the Seas of the USSR, 1992).

Barotropic and baroclinic free-sea level oscillations with periods of 13–41 h represent multi-node progressive-standing waves with amphidromic systems rotating in different directions. The speed of the isophase rotation in barotropic amphidromic systems of 13 and 27 h periods is close to the theoretical phase speed of barotropic



gravity waves, while the phase speed of the amphidromic systems with a 41 h period is lower than that of gravity
waves. In baroclinic conditions, the values of PSW phase speeds usually disagree with the theoretical values of
GW phase speed estimated for stratified sea.
Correct identification of the described free barotropic and baroclinic oscillations in the Baltic Sea can help to
explain many large-scale variabilities of different physical characteristics (for example, large-scale sea level
changes).
According to theoretical investigations by LeBlond and Mysak (1978), a sea basin is characterised by its own set
of frequencies of barotropic and baroclinic oscillations. These oscillations refer to two classes. The
eigenoscillations of the first class are long gravity waves representing longitudinal waves. In no-boundary ocean
conditions and under the effect of the Earth's rotation, this type of wave is generated with frequencies that are
above a local inertial frequency. An introduction of a boundary results in trapping the wave energy and
generating trapped gravity Kelvin waves (Pedlosky, 1979). The Kelvin wave is the only wave type existing in
both band frequencies, above and below the inertial frequency (Efimov et al., 1985). Kelvin waves always
propagate anticlockwise in the Northern Hemisphere and clockwise in the Southern Hemisphere.
Eigenoscillations of the second class are planetary waves. Among them, Rossby and topographic waves have
been extensively investigated (LeBlond and Mysak, 1978). Rossby waves are horizontal transverse waves that
are generated in the frequency band, which are below inertia frequencies (Pedlosky, 1979). Rossby waves
always propagate westward, while topographic waves move along isobath lines and leave sharp bathymetry from
their right in the Northern Hemisphere and from their left in the Southern Hemisphere.
In semi-enclosed sea basins, the mechanism of wave reflection may have an significant effect on the propagation
of long waves and can lead to the generation of progressive-standing modes of gravity and planetary waves
(Nekrasov et al., 1975; LeBlond and Mysak, 1978; Pedlosky, 1979).
An earlier theoretical investigation of the dynamics of topographic Rossby waves in enclosed basins (Buchwald,
1973; LeBlond and Mysak, 1978; Pedlosky, 1979) demonstrated that they may have characteristics both standing
and progressive waves. Two types of node lines were observed in the Longuet-Higgins (1965) study during the
experiment in a rectangular basin: lines approximated by an envelope function with nodes stable in spatio-
temporal domain, as well as lines of progressive waves moving westward with a Rossby wave phase speed.
Theoretical studies of long gravity waves in enclosed or semi-enclosed basins that account for the Earth's
rotation have shown that these waves transform into multi-node progressive-standing Kelvin waves (Taylor,
1922; Nekrosov, 1975; Pugh, 1987). The overall effect of the Earth's rotation on free oscillations is to vitiate the
development of fixed nodal lines and to atrophy them into nodal points or amphidromic centres (Wilson, 1972).
Then, an oscillation rotates around the amphidromic centre in the form of a Kelvin wave such that the amplitude
decreases from zero in its centre to the maximum on basin boundaries. In the Northern Hemisphere, this rotation
is anticlockwise and changes to clockwise in the Southern Hemisphere.
Besides the Coriolis force, the opposite rotation of isophases in amphidromic systems may result from the
interference of standing waves (Harris, 1904, Proudman, 1953, Nekrasov, 1975, Schwiderski, 1979). Multiple
combinations of amplitude, angle, and phase differences of interfering waves are possible and may lead
anticlockwise to clockwise rotation.
The analysis of our numerical simulations coincides with the results of these theoretical experiments. The
opposite phase rotation is found for PSWs with a 27 h period (anticlockwise, similar to the Kelvin wave) and for





PSWs with 13 h and 41 h periods (clockwise). The comparison of the phase speed of simulated free barotropic
oscillations with theoretical values suggests that most of the oscillations with periods of 13 h and 27 h are
barotropic gravity waves. Other waves with 27 h periods and almost all waves with 41 h periods are likely to be
related to barotropic modes of topographic Rossby waves as their phase speed is lower than that of the
theoretical barotropic gravity waves, and their period is longer than that of inertial oscillations.
Compared with barotropic conditions, the number and location of amphidromic systems in a stratified sea
change remarkably (Fig. 7a and Fig. 12a). By their phase speed, most of the free oscillations in the baroclinic
conditions of high (13 h) and medium (27 h) frequencies, as well as a substantial portion of oscillations at a
lower frequency (41 h), can be identified as barotropic gravity waves.
Our experiments demonstrate that in a stratified sea, the percentage of relatively slow-moving free waves
significantly increases compared with an unstratified sea. These changes can be associated with the generation of
the baroclinic mode of the topographic Rossby waves in a stratified medium. The significant difference in the
phase pattern in baroclinic and barotropic conditions (see Section 3.2) can be explained by the superposition of
the phases of 1) barotropic gravity waves and 2) barotropic/baroclinic modes of the topographic Rossby waves.
We also noted that there is no evidence of the existence of the baroclinic mode of long gravity waves in the
Baltic Sea because most of our phase speed estimates for the 13 and 27 h oscillations do not agree with the range
of theoretical phase speeds of the internal gravity waves estimated for local baroclinic conditions.
Free sea level oscillations at seasonal scales (periods of 3 months to 1 year) have a baroclinic origin, as they
appear only in baroclinic simulations. The phase speed of the oscillations in the 358-day period is lower than the
theoretical values for the internal gravity waves and significantly varies from the range of values typical for
barotropic gravity waves. We relate these oscillations to the baroclinic mode of topographic Rossby waves. A
fraction of waves of the 89-day period is also the topographic Rossby waves. However, the other part of these
oscillations has phase speeds (0.01–0.24 m s$^{-1}$) overlapping with the range of theoretical values of the internal
gravity waves (0.08–1.53 m s$^{-1}$). This part can be identified as a baroclinic gravity wave.
Several studies have demonstrated that amplitudes of seasonal fluctuations in the Baltic sea level have important
inter-annual variability (Ekman, 1998; Stramska et al., 2013; Barbosa and Donner, 2016; Cheng et al., 2018).
Considering that the free oscillations of seasonal-scale frequencies have a baroclinic origin, we hypothesise that
they could contribute to the non-stationary nature of these seasonal fluctuations. Major Baltic Inflows (MBIs) are
well-known sporadic events that import saline waters into the Baltic. In recent decades, their occurrence has
changed significantly (Fischer and Matthaus, 1996; Matthaus, 2006). The MBI events, along with the inter-
annual variability of the freshwater input via atmospheric precipitation and river flow affect the Baltic Sea water
mass stratification (Assessment of Climate, 2008). The inter-annual variability in the stratification, in turn, may
affect the frequencies of the baroclinic modes of the Baltic Sea eigenoscillations. As a result, from year to year,
the resonance of atmospheric forces with the baroclinic modes of free sea level oscillations can occur at different
seasonal-scale frequencies or may not occur at all. This mechanism could be one of the reasons responsible for
the unsteady character of the Baltic sea level seasonal variability and will be studied in the future.

**Conclusion**





The results of our numerical simulations of the free sea level oscillations of the Baltic Sea revealed a general similarity, with a distinct difference in the processes of relaxation of sea level oscillation in barotropic and baroclinic conditions.

1. The predominant common feature is the generation of oscillations in the same mesoscale frequency range (13–41 h) in both the unstratified and stratified sea experiments. These oscillations have the form of one- or multi-node progressive-standing waves with amphidromic systems rotating in opposite directions depending on the oscillation period.

2. The primary difference is the generation of sea level baroclinic oscillations at seasonal scales with periods of 89 and 358 days.

**3.** The highest amplitudes of free barotropic oscillations occur at the top of the Gulf of Finland, the Gulf of Bothnia, in the south-western Baltic Proper, and in the Kalmar Strait. The highest amplitudes of baroclinic oscillations are found in the deep areas with the highest stratification of water masses in the Baltic Proper.

4. Free barotropic oscillations of periods of 13 h and 27 h represent long gravity waves. Most of the 41 h period barotropic oscillations are likely to be the barotropic mode of the topographic Rossby wave.

5. The essential part of free oscillations of 13–41 h periods in the baroclinic conditions may be regarded as topographic Rossby waves generated in semi-enclosed basins. However, there is a minor part of these oscillations that represent barotropic gravity waves. We did not find evidence of the existence of the baroclinic mode of long gravity waves at these frequencies.

6. Regarding free oscillations at a seasonal scale, we suggest that all oscillations of 358 days and half of the oscillations of 89 days are related to the baroclinic mode of the topographic Rossby waves, as their phase speeds do not overlap with the theoretical values estimated for internal gravity waves. However, the other part of 89-day baroclinic oscillations, with their phase speed, is likely to be the baroclinic gravity waves.

Based on the results of our numerical experiments, we can conclude that after the cessation of the atmospheric forcing, the relaxation of the Baltic free sea level oscillations occurs in the form of barotropic and baroclinic modes of progressive-standing gravity waves as well as in the form of topographic Rossby waves. The free baroclinic oscillations contribute significantly to the spectre of the Baltic Sea eigenoscillations. Their role is the most important in seasonal-scale sea level fluctuations.

**Acknowledgements**. This research was made possible with support from  the Saint-Petersburg University, Grant № IAS_18.37.140.2014. The authors express their gratitude towards Dr. A.Kouraev (Toulouse University) for his valuable support and suggestions.

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
