# Peer review of "Spatio-temporal structure of Baltic free sea level oscillations 1 in barotropic and baroclinic conditions from hydrodynamic 2 modelling. 3"

_Ocean Science, 2020_

## Referee Comment (RC1) · Anonymous Referee #1 · 23 Dec 2020

This is an interesting piece of research that sheds more light on the properties of barotropic and baroclinic self-oscillations of the Baltic Sea using a contemporary 3D ocean model and standard means of decomposition of velocity fields and sea level time series into Fourier components.

The description of the background and the research problem is professional. I would only recommend to (i) include reference to a generic overview of physical oceanography of the Baltic Sea (Leppäranta and Myrberg, 2009) already in the Introduction, (ii) mention an early but deep and still actual overview of the problem [Samuelsson,

[Figure]

M., Stigebrandt, A. Main characteristics of the long-term sea level variability in the Baltic Sea. Tellus A, 48(5), 672-683, doi: 10.1034/j.1600-0870.1996.t01-4-00006.x, 1996], and (iii) discuss the results for the baroclinic calculations, if applicable, in the light of some recent advances towards better understanding of low-frequency baroclinic oscillations of the Baltic Sea [Kurkin, A. et al. Spatial distribution of energy of subinertial baroclinic motions in the Baltic Sea, Frontiers in Earth Science, 8, 184, doi: 10.3389/feart.2020.00184, 2020], with recommended focus on material on lines 398–407.

The hydrodynamic model INMOM is professional and up-to-date (although not much used as the input for international research publications). The idea is to spin up the system for a certain time interval under the impact of a strong atmospheric forcing, and then let the system run at its own, with the goal to detect the maximum number of self-oscillations over two years. To do so, the authors implement several simplifications that are not fully realistic but eventually help to detect various kinds of self-oscillations. In essence, the system of detected oscillations should be invariant with respect to the initial disturbance; however, this feature should be at least shortly discussed (and at best with some supporting evidence from other parts of the World). Also, in this context it would be important to explain why in both the barotropic and baroclinic implementations, the Baltic Sea was considered a fully enclosed basin, with no water exchange with the North Sea as stated on lines 175–177. It is also important to comment shortly on possible differences with runs that would resolve water exchange between the North and Baltic Sea. Also, it remains unclear whether river water input and ice conditions were also neglected in both the barotropic and baroclinic implementations (line 177).

On lines 182–183 it is said that "Setting the turbulent viscosity to zero for the vertical components and to the minimum values for the horizontal components allows the damping of the simulated sea level fluctuations to be reduced." Please comment whether you did so and, if yes, how the modelled spectra relate to the actual spectra of motions. Again, I guess, the results are qualitatively invariant with respect to the particular set of settings, and stronger damping would simply render some self-oscillation patterns undetectable.

The text is written in good English, with a few minor items to adjust. I recommend the manuscript for publication with minor and fairly straightforward revisions along the recommendations above and below.

Minor aspects to adjust:

Line 29: "may resonate" would be more exact.

Lines 30–31: The source "Kulikov and Medvedev, 2013" addresses generic spectrum of water level in the Baltic Sea and seems inappropriate in the context of this particular claim.

Lines 37–39: For the benefit of readers I recommend to include also a reference to (Leppäranta and Myrberg, 2009). The provided references are correct but they are not easily accessible today, and some of them are written in German or Russian.

Line 41: "a 39 h period".

Line 42: must be "Neumann".

Line 60 and elsewhere: I recommend using "Baltic proper" as this is not a proper name.

Lines 87–88: it is recommended to say already here that the model uses sigma-coordinates in vertical. Line 104: better say "with a spatial resolution of 2 nautical miles" or similar, and indicate the resolution also in kilometers (cf. line 115).

Line 113: capitalize: "HIROMB".

Lines 116, 187: capitalize: "ERA".

Lines 133–134: remove repeating information "and includes the station coordinates, sea level measurement frequency, number of sea level measurements used in this study, and percentage of missing data" that is found in the table caption.
**OSD**

Line 138, Table 1: (i) remove the column "Period" and mention this interval in the caption; (ii) use "measurement interval/frequency" or similar instead of "span".

Line 143: better say "standard statistical parameters".

Line 144: define SSH for the reader.

Line 144: obviously "ratio" is meant for $\sigma_p$.

Line 145: the additional criteria of accuracy $P_{m}$ is not defined but still used below (e.g., on line 166). Lines 146–157: please punctuate the text in formulas as part of the sentences.

Lines 152–154: I don't think it makes sense to show the basic formulas (3)–(5) in a research paper of this type.

Line 166: please check the format the expression "Pm < 0.674$\sigma$tg".

Line 171: "again" is redundant.

Line 175: should be "exchange".

Line 199: the expression "k=0,1,2..." should be removed as the role of k becomes evident from the expression for f(t); also, the definition of angular frequency would be better placed in line with the rest or the text.

Line 201: please check the format a_k.

Line 202: "mean average" and "coefficient number" sound strange.

Line 203: please check the format F_k and A_k.

Line 203: should be Equation (8).

Lines 207, 211: why period is now P? Does it have a specific meaning?

Lines 208–214: please punctuate the text in formulas as part of the sentences and check the format of variables in the text.

[Figure]

Line 220: the words "are low and" are redundant.

Line 222: "principally" is redundant.

Line 229: The standard deviation appears now as ($\sigma$m); please redefine or unify with the usage above.

Line 232, 234: please show "Hailuoto Island, Ulvö Deep" etc. on some map.

Line 310: should be "Leppäranta and Myrberg, 2009)."

Line 311–312: include the classic expression for the long wave speed into the text. There is no need for a displayed equation.

Line 322: probably "phenomena" or similar are meant (instead of "observations).

Line 336: show Narva Bay and Ellesmere Island on some map.

Line 349: "the pycnocline".

Line 350: say "barotropic oscillations".

Line 351: simply "broaden".

Line 360: show Vyborg Bay on some map.

Line 378: include the classic expression for the long wave speed into the text also here. There is no need for a displayed equation.

Line 380, Figure 13: say "Number of cases" in the legend ($2\times$).

Lines 386–387: say simply "we estimated the phase speed ...".

Line 413, Figure 15: say "Number of cases" in the legend ($2\times$).

Line 416: remove "estimated by (Eq. 13)".

Line 423, the expression "the maximum dispersion of free oscillations occurs" seems to contain too much cryptic jargon. Consider saying: "the most intense free oscillations

occur" or similar.

Line 429: consider replacing "moves to" by "is located in".

Line 436, 474, 475: it is recommended to use the long expression instead of PSW and GW in this sort of discussion as some readers may omit the body part of the paper.

Line 445: use "the local".

Line 464: should be (Nekrasov, 1975).

Line 504: (Fischer and Matthaus, 1996; Matthaus, 2006) missing from the reference list; also, should be : (Fischer and Matthäus, 1996; Matthäus, 2006).

Lines 521–522: please put the claim into more clear connection with the properties of barotropic oscillations.

Line 523: consider replacing "at the top" by some other expression.

Line 532: "may represent" seems to be more accurate.

References: (i) most doi indices are missing; please amend; (ii) the style of most references does not follow the Ocean Science style.

Line 606: should be "Longuet-".

Line 608: the title should be capitalized according to German style: "Spektren der Wasserstandsschwankungen der Ostsee im Jahre 1958".

Line 618: should be "Suursaar".

Line 632: delete most of the reference and leave only: "Wilson B. W. Seiches. Advances in Hydroscience, 8, 1–89, 1972."

---

## Referee Comment (RC2) · Anonymous Referee #2 · 4 Jan 2021

General comments

In the paper, 3D hydrodynamic model simulations are used to study the free relaxation oscillations that would occur in the Baltic Sea once the meteorological forcing ceases. This is an interesting theoretical analysis of the Baltic Sea level behaviour. Corresponding analyses on the eigenoscillations of the Baltic Sea have mainly been conducted with simpler (barotropic) models, many of them several decades ago. Thus, this new study is a nice addition to the topic.

[Figure]

The paper is well organised and clearly written. The hydrodynamic model and the methods used for the analyses are well presented. The results are also thoroughly and clearly presented. The conclusions on the nature of the different oscillations form an interesting discussion. I only suggest some minor clarifications, see comments below.

Specific comments

The oscillations studied in this paper occurred in a model where some parameters were adjusted to unrealistic values in order to reduce damping (l. 182-183). It would be interesting to see some discussion on how the results obtained relate to the sea level behaviour in the real Baltic Sea. Is there a possibility that the parameter adjustments affect the oscillation frequencies? How much are such oscillations expected to contribute to the real sea level variability? How fast would they be damped?

Fig. 8: Why is there so much white space in these maps? The areas around the oscillation nodes are apparently excluded due to low amplitude. But why are e.g. phase speeds for the eastern Gulf of Finland missing in Fig. 8b, even if the amplitude of the oscillation should be high (Fig. 7b)?

In a seasonal scale in the baroclinic simulation, after all the external forcing ceases, I would assume that something happens to the temperature and salinity distribution also. Were such processes considered, and how would they affect the surface height?

l. 499-502. Most of the interannual variability in the seasonal sea level fluctuations likely originates directly from the interannual variability in the atmospheric forcing. E.g. the role of the air pressure conditions, the NAO index, etc., have been shown to explain a significant portion of the interannual variability. Thus, I suppose the contribution from the baroclinic free oscillations is minor. (Which might be mentioned.)

Technical corrections

l. 41 and elsewhere: "Bothnia Bay" => Please use either "Bothnian Bay" or "Bay of Bothnia" consistently.

l. 42: Newman => Neumann.

l. 144: "observed (sigma_m) and simulated (sigma_tg)" => "observed (sigma_tg) and simulated (sigma_m)".

l. 144-145: sigma_p is not the relation of sigma_m and sigma_tg (as stated), but sigma_er and sigma_tg.

l. 145-146: Please give the definition of P_m. Now it remains unclear which measure should be <0.674*sigma_tg.

l. 201: ...a_k and b_k are the coefficients...

l. 202: "mean average" => "mean" or "average"

l. 207-211: Is the period P same as T above? If so, please use the same symbol. If not, please explain.

Fig. 4: The original amplitude of the displacement ranges from -50 to +100 cm in Fig. 3. How does this relate to the plots in Fig. 4 starting from around +10-20 cm at every station? What is the vertical axis in these?

Fig. 4: It would ease the comparison of amplitudes if all subplots had the same y axis (as they are very close to each other already).

l. 229 and Fig. 5: What is "standard deviation of amplitudes"? If this is the standard deviation of time series, as Eq. (4) implies, then the word "amplitude" here is misleading.

l. 230: Please check the font size of all subscripts, here and elsewhere.

l. 233: Consider adding locations of Pärnu Bay and Rügen Island to Fig. 1.

l. 235-236: "Oscillations of medium intensity can be noted as over local uplifts in the Baltic Proper as over-bottom depressions..." Please reformulate this sentence.

l. 261: There is a 17-h period listed. However, there are no 17-h peaks in Fig. 6. This

should be 16-h?

l. 310: Leppäranta and Myrberg, 2009

l. 311: From Eq. (13), a range of 12-67 m/s corresponds to depths of 15-458 m. From where does the lowest limit of 15 m come? Please specify.

l. 321: "anemobaric forcing": please specify what is meant by this. Is "anemobaric" a synonym to the entire meteorological forcing described on lines 118-119?

l. 322: "30-35 cm in amplitude" => "in range"? In Fig. 9, the largest range of variations seems to be 35 cm. This is not amplitude, which by definition is half of the total range.

l. 330: "standard deviation of the amplitudes"; see comment above.

l. 336: Ellesmere Island? Please check the name, and add location to Fig. 1.

l. 362: "Free oscillations of 27 h periods in the baroclinic conditions reached the maximum in the narrow zone near the southwest Finland coast". There seems to be a much more apparent maximum in the eastern Gulf of Finland in Fig. 12b, please check this sentence.

l. 382-383: "Dash line on the histogram plots indicates minimum theoretical value of phase speed of baroclinic (Ci) and barotropic (Cg) gravity waves." It looks like the dash line for Ci indicates the maximum value (1.53 m/s), not minimum.

l. 412: May be more specific here: "significantly lower for 358-day waves and belong to the theoretical range for 89-day waves". I see this is what is meant, but "longer" and "shorter" are a bit too generic and it is hard to understand the sentence.

Fig. 15. Line 388 says the Ci range is 0.08-1.53 m/s. Why is the maximum lower here?

l. 467: decreases => increases

l. 479: It would be helpful to mention explicitly the period of inertial oscillations in the area (about 14 hours).

**[OSD](https://doi.org/10.5194/os-2020-110)**

---

## Author Comment (AC1) · 27 Jan 2021

We appreciate the Referee valuable remarks and recommendations and carefully addressed them in the new version of the manuscript. Our answers on the major comments can be found in the text bellow and full answers are given in the Supplement materials.

On behalf of all authors,

Elena Zakharova

[Figure]

Referee 1 major comment

The description of the background and the research problem is professional. I would only recommend to (i) include reference to a generic overview of physical oceanography of the Baltic Sea (Leppäranta and Myrberg, 2009) already in the Introduction, (ii) mention an early but deep and still actual overview of the problem [Samuelsson, M., Stigebrandt, A. Main characteristics of the long-term sea level variability in the Baltic Sea. Tellus A, 48(5), 672-683, doi: 10.1034/j.1600-0870.1996.t01-4-00006.x, 1996], and (iii) discuss the results for the baroclinic calculations, if applicable, in the light of some recent advances towards better understanding of low-frequency baroclinic oscillations of the Baltic Sea [Kurkin, A. et al. Spatial distribution of energy of subinertial baroclinic motions in the Baltic Sea, Frontiers in Earth Science, 8, 184, doi: 10.3389/feart.2020.00184, 2020], with recommended focus on material on lines 398–407.

Reply: Thank you. We included the references to the Leppäranta and Myrberg, (2009); and Samuelsson and Stigebrandt, (1996) overviews as recommnded (see page1). However, Kurkin, et al,( 2020) does not investigate the water level variability in baroclinic waves, therefor the reference to this publication would not be fully correct.

The hydrodynamic model INMOM is professional and up-to-date (although not much used as the input for international research publications). The idea is to spin up the system for a certain time interval under the impact of a strong atmospheric forcing, and then let the system run at its own, with the goal to detect the maximum number of self-oscillations over two years. To do so, the authors implement several simplifications that are not fully realistic but eventually help to detect various kinds of self-oscillations. In essence, the system of detected oscillations should be invariant with respect to the initial disturbance; however, this feature should be at least shortly discussed (and at best with some supporting evidence from other parts of the World).

Reply: 1. Thank you for this feedback. We agree with the Reviewer that in our definition

of task, simulated free sea level oscillations should be invariant relative to initial per-turbation. Moreover, the agreement of our results with the results of cited publications proves this statement. In spite of difference of forcing conditions used in different stud-ies, the resulting spectra of free barotropic oscillations are quite similar. Unfortunately, we have not found publications for other World, where the baroclinic modes of free oscillations were investigated using numerical experiments based on hydrodynamic model and were not able to discuss this question.

Also, in this context it would be important to explain why in both the barotropic and baro-clinic implementations, the Baltic Sea was considered a fully enclosed basin, with no water exchange with the North Sea as stated on lines 175–177. It is also important to comment shortly on possible differences with runs that would resolve water exchange between the North and Baltic Sea. Also, it remains unclear whether river water input and ice conditions were also neglected in both the barotropic and baroclinic implemen-tations (line 177).

Reply: We introduce requested details to clarify this remark (see section 2.3). 1."This assumption aimed to exclude the effect of external barotropic and baroclinic oscilla-tions coming from the North Sea". 2."River water input and ice conditions were also neglected in both numerical experiments."

On lines 182–183 it is said that "Setting the turbulent viscosity to zero for the ver-tical components and to the minimum values for the horizontal components allows the damping of the simulated sea level fluctuations to be reduced." Please comment whether you did so and, if yes, how the modelled spectra relate to the actual spectra of motions. Again, I guess, the results are qualitatively invariant with respect to the par-ticular set of settings, and stronger damping would simply render some self-oscillation patterns undetectable.

Reply. We agree with the remark and edited the text for better reading: "In order to be able to characterize the free oscillations with better spectral resolution and in larger

spectral range, the sea level dumping factors have to be reduced. In both numerical experiments, the dumping effect was reduced due to 1) setting the coefficients of vertical turbulent viscosity and of bottom friction to zero and 2) setting the coefficient of horizontal turbulent viscosity to the minimum values."

Please also note the supplement to this comment:
https://os.copernicus.org/preprints/os-2020-110/os-2020-110-AC1-supplement.pdf

**Supplement:**

We appreciate the Referee valuable remarks and recommendations and carefully addressed them in the new version of the manuscript. Our answers on the major comments can be found in the text bellow and full answers are given in the Supplement materials.

**On behalf of all authors,**

**Elena Zakharova**

Referee 1 major comment

The description of the background and the research problem is professional. I would only recommend to (i) include reference to a generic overview of physical oceanography of the Baltic Sea (Leppäranta and Myrberg, 2009) already in the Introduction, (ii) mention an early but deep and still actual overview of the problem [Samuelsson, M., Stigebrandt, A. Main characteristics of the long-term sea level variability in the Baltic Sea. Tellus A, 48(5), 672-683, doi: 10.1034/j.1600-0870.1996.t01-4-00006.x, 1996], and (iii) discuss the results for the baroclinic calculations, if applicable, in the light of some recent advances towards better understanding of low-frequency baroclinic oscillations of the Baltic Sea [Kurkin, A. et al. Spatial distribution of energy of subinertial baroclinic motions in the Baltic Sea, Frontiers in Earth Science, 8, 184, doi: 10.3389/feart.2020.00184, 2020], with recommended focus on material on lines 398–407.

**Reply: Thank you. We included the references to the Leppäranta and Myrberg, (2009); and Samuelsson and Stigebrandt, (1996) overviews as recommnded (see page1). However, Kurkin, et al,( 2020) does not investigate the water level variability in baroclinic waves, therefor the reference to this publication would not be fully correct.**

The hydrodynamic model INMOM is professional and up-to-date (although not much used as the input for international research publications). The idea is to spin up the system for a certain time interval under the impact of a strong atmospheric forcing, and then let the system run at its own, with the goal to detect the maximum number of self-oscillations over two years. To do so, the authors implement several simplifications that are not fully realistic but eventually help to detect various kinds of self-oscillations. In essence, the system of detected oscillations should be invariant with respect to the initial disturbance; however, this feature should be at least shortly discussed (and at best with some supporting evidence from other parts of the World).

**Reply: 1. Thank you for this feedback. We agree with the Reviewer that in our definition of task, simulated free sea level oscillations should be invariant relative to initial perturbation. Moreover, the agreement of our results with the results of cited publications proves this statement. In spite of difference of forcing conditions used in different studies, the resulting spectra of free barotropic oscillations are quite similar. Unfortunately, we have not found publications for other World, where the baroclinic modes of free oscillations were investigated using numerical experiments based on hydrodynamic model and were not able to discuss this question.**

Also, in this context it would be important to explain why in both the barotropic and baroclinic implementations, the Baltic Sea was considered a fully enclosed basin, with no water exchange with the North Sea as stated on lines 175–177. It is also important to comment shortly on possible differences with runs that would resolve water exchange between the North and Baltic Sea. Also, it remains unclear whether river water input and ice conditions were also neglected in both the barotropic and baroclinic implementations (line 177).

**Reply: We introduce requested details to clarify this remark (see section 2.3).**

**1."This assumption aimed to exclude the effect of external barotropic and baroclinic oscillations coming from the North Sea".**
**2."River water input and ice conditions were also neglected in both numerical experiments."**

On lines 182–183 it is said that "Setting the turbulent viscosity to zero for the vertical components and to the minimum values for the horizontal components allows the damping of the simulated sea level fluctuations to be reduced." Please comment whether you did so and, if yes, how the modelled spectra relate to the actual spectra of motions. Again, I guess, the results are qualitatively invariant with respect to the particular set of settings, and stronger damping would simply render some self-oscillation patterns undetectable.

**Reply. We agree with the remark and edited the text for better reading:**
**"In order to be able to characterize the free oscillations with better spectral resolution and in larger spectral range, the sea level dumping factors have to be reduced. In both numerical experiments, the dumping effect was reduced due to 1) setting the coefficients of vertical turbulent viscosity and of bottom friction to zero and 2) setting the coefficient of horizontal turbulent viscosity to the minimum values."**

Minor aspects to adjust:

Line 29: "may resonate" would be more exact.
**The phrase was changes according to Reviewer suggestion on "may resonate"**

Lines 30–31: The source "Kulikov and Medvedev, 2013" addresses generic spectrum of water level in the Baltic Sea and seems inappropriate in the context of this particular claim.
**Thank you for catching this error. The reference was removed from this phrase.**

Lines 37–39: For the benefit of readers I recommend to include also a reference to (Leppäranta and Myrberg, 2009). The provided references are correct but they are not easily accessible today, and some of them are written in German or Russian.
**The reference was added**

Line 41: "a 39 h period".
**Thank you, the text is edited**

Line 42: must be "Neumann".
**Thank you, the text is edited**

Line 60 and elsewhere: I recommend using "Baltic proper" as this is not a proper name.
**The text was edited accordint to this suggestion.**

Lines 87–88: it is recommended to say already here that the model uses sigma coordinates in vertical.
**In our text the sigma is defined already in the second sentence dedicated to the model description. " A dimensionless value $\sigma$ is used as the vertical coordinate...". If the comment has another meaning, sorry, we did not catch it.**

Line 104: better say "with a spatial resolution of 2 nautical miles" or similar, and indicate the resolution also in kilometers (cf. line 115).
**Thank you for suggestion, the text was edited**

Line 113: capitalize: "HIROMB".
**The text was edited**

Lines 116, 187: capitalize: "ERA".
**The text was edited**

Lines 133–134: remove repeating information "and includes the station coordinates, sea level measurement frequency, number of sea level measurements used in this study, and percentage of missing data" that is found in the table caption.
**The sentens was edited as recommended.**

Line 138, Table 1: (i) remove the column "Period" and mention this interval in the caption; (ii) use "measurement interval/frequency" or similar instead of "span".
**The corrections were introduced**

Line 143: better say "standard statistical parameters".
**The correction was introduced**

Line 144: define SSH for the reader.
**The correction was introduced**

Line 144: obviously "ratio" is meant for $\sigma_p$.
**The correction was introduced**

Line 145: the additional criteria of accuracy $P_{m}$ is not defined but still used below (e.g., on line 166).
**The definition of $P_m$ was given in the next sentence, but, indeed, was poorly worded. The text was rewritten : "The $P_m$ parameter allows the assessment of the number of good simulations considering their accuracy $< 0.674\sigma_{tg}$. "**

Lines 146–157: please punctuate the text in formulas as part of the sentences.
**The commas were added.**

Lines 152–154: I don't think it makes sense to show the basic formulas (3)–(5) in a research paper of this type.
**According to recommendations equations (3-5) were deleted.**

Line 166: please check the format the expression "Pm < 0.674σtg".
**Thank you for catching this error. The correction was introduced**

Line 171: "again" is redundant.
**The word "again" was removed.**

Line 175: should be "exchange".
**The correction was introduced**

Line 199: the expression "k=0,1,2. . ." should be removed as the role of k becomes evident from the expression for f(t); also, the definition of angular frequency would be better placed in line with the rest or the text.
**The corrections were introduced**

Line 201: please check the format a_k.
**The correction was introduced**

Line 202: "mean average" and "coefficient number" sound strange.
**The correction was introduced**

Line 203: please check the format F_k and A_k.
**The correction was introduced**

Line 203: should be Equation (8).
**The correction was introduced**

Lines 207, 211: why period is now P? Does it have a specific meaning?
**The correction was introduced: P was changed for T in the text and corresponding equations.**

Lines 208–214: please punctuate the text in formulas as part of the sentences and check the format of variables in the text.
**The corrections were introduced**

Line 220: the words "are low and" are redundant.
**The correction was introduced: the words were deleted.**

Line 222: "principally" is redundant.
**The word was deleted.**

Line 229: The standard deviation appears now as (σm); please redefine or unify with the usage above.
**The correction was introduced.**

Line 232, 234: please show "Hailuoto Island, Ulvö Deep" etc. on some map.
**The island and the Deep are snown now on the fig.1 map.**

Line 310: should be "Leppäranta and Myrberg, 2009)."
**The correction was introduced.**

Line 311–312: include the classic expression for the long wave speed into the text. There is no need for a displayed equation.
**The Equation (13) was moved into the text. " Under these conditions, the theoretical phase speed of the barotropic gravity wave in the Baltic Sea, calculated using expression** $C_g = \sqrt{gH}$ **(where $H$ is the depth and $g$ is the acceleration due to gravity), ranges between 12 and 67 m s$^{-1}$. "**

Line 322: probably "phenomena" or similar are meant (instead of "observations).
**Thank you for catching this error. The "observations" was changed for "phenomena".**

Line 336: show Narva Bay and Ellesmere Island on some map.
**The Narva Bay is now shown on the map. The Ellesmere Island has to be Öland Island. Necessary correction was made.**

Line 349: "the pycnocline".
**The correction was introduced.**

Line 350: say "barotropic oscillations".
**The correction was introduced.**

Line 351: simply "broaden".
**The correction was introduced.**

Line 360: show Vyborg Bay on some map.
The Vyborg Bay is now shown on the Fig1.

Line 378: include the classic expression for the long wave speed into the text also here. There is no need for a displayed equation.
**The correction was introduced.**

Line 380, Figure 13: say "Number of cases" in the legend (2×).
**The correction was introduced.**

Lines 386–387: say simply "we estimated the phase speed . . .".
**The phrase was simplified according to recommendations.**

Line 413, Figure 15: say "Number of cases" in the legend (2×).
**The correction was introduced.**

Line 416: remove "estimated by (Eq. 13)".
**The correction was introduced.**

Line 423, the expression "the maximum dispersion of free oscillations occurs" seems to contain too much cryptic jargon. Consider saying: "the most intense free oscillations occur" or similar.
**The correction was introduced. " In barotropic conditions, the most intense free oscillations occur on a time scale of...."**

Line 429: consider replacing "moves to" by "is located in".
**The correction was introduced.**

Line 436, 474, 475: it is recommended to use the long expression instead of PSW and GW in this sort of discussion as some readers may omit the body part of the paper.
**In the discussion section the PSW and GW abbreviations were replaces for long expressions.**

Line 445: use "the local".
**The correction was introduced.**

Line 464: should be (Nekrasov, 1975).
**The correction was introduced.**

Line 504: (Fischer and Matthaus, 1996; Matthaus, 2006) missing from the reference list; also, should be : (Fischer and Matthäus, 1996; Matthäus, 2006).
**The references were added and corrections were introduced.**

Lines 521–522: please put the claim into more clear connection with the properties of barotropic oscillations.
**The phrase was rewritten. " The primary difference between the results of the experiments consists in the generation of sea level baroclinic oscillations of seasonal scales with periods of 89 and 358 days in stratified sea".**

Line 523: consider replacing "at the top" by some other expression.
**Replaced for "in the eastern part of the Gulf of Finland"**

Line 532: "may represent" seems to be more accurate.
**The correction was introduced.**

References: (i) most doi indices are missing; please amend; (ii) the style of mostreferences does not follow the Ocean Science style.
**The reference list and format was edited with the Mendeley to match with the Copernicus publishing format.**

Line 606: should be "Longuet-".
**The correction was introduced.**

Line 608: the title should be capitalized according to German style: "Spektren der Wasserstandsschwankungen der Ostsee im Jahre 1958".
**The correction was introduced.**

Line 618: should be "Suursaar".
**The reference on this article does not exist in body text and erroneously appears in the Reference list. The correction was made.**

Line 632: delete most of the reference and leave only: "Wilson B. W. Seiches. Advances in Hydroscience, 8, 1–89, 1972."
**The correction was introduced.**

---

## Author Comment (AC2) · 27 Jan 2021

We appreciate the Referee valuable remarks, recommendations and his profound corrections and carefully addressed them in the new version of the manuscript. . Our answers on the major comments can be found in the text bellow and full answers are given in the Supplement materials.

On behalf of all authors,

Elena Zakharova

[Figure]

Referee 2 major comment.

The oscillations studied in this paper occurred in a model where some parameters were adjusted to unrealistic values in order to reduce damping (l. 182-183). It would be interesting to see some discussion on how the results obtained relate to the sea level behaviour in the real Baltic Sea. How much are such oscillations expected to contribute to the real sea level variability? Is there a possibility that the parameter adjustments affect the oscillation frequencies?

Reply: 1.The parameters of adjustments affect the dissipation speed and do not affect the frequencies. If we do not reduce these parameters we can't catch the oscillations of seasonal scale. We edited corresponding phrase in the text for clarification. 2. A comparison of tide gauge and numerical simulation spectra is shown on figure XX . The real sea level fluctuation is the superposition of forced and free oscillations of different origin. The tide gauge spectrum contains very big amount of peaks those amplitudes are significantly higher. In contrast, the spectrum of simulated free oscillations is characterised by a small number of peaks of lower amplitudes which are masked by forced oscillations. So, we decided to exclude comparison of real and simulated spectra from the discussion.

How fast would they be damped?

Reply: The answer for this question is given in the Section 2.3. "Under natural conditions, the free sea level oscillations attenuate rapidly due to the dissipative effects of vertical and horizontal viscosity, near-bottom friction, non-linear effects, and Earth's rotation (Proshutinsky 1993, Zakharchuk et al., 2004). According to a theoretical concept and previous numerical experiments (Proudman, 1953; Wübber and Kraus, 1979; Zakharchuk et al., 2004; Leppäranta and Myrberg, 2009), the relaxation of the Baltic large-scale free sea level oscillations takes several days."

Fig. 8: Why is there so much white space in these maps? The areas around the oscillation nodes are apparently excluded due to low amplitude. But why are e.g. phase

speeds for the eastern Gulf of Finland missing in Fig. 8b, even if the amplitude of the oscillation should be high (Fig. 7b)?

Reply: We introduced additional phrase to the caption of corresponding figures and following phrase into the body text. "In areas where $\Delta Fx$ and $\Delta Fy$ equal to zero (white areas on fig 8), the standing wave component prevails."

In a seasonal scale in the baroclinic simulation, after all the external forcing ceases, I would assume that something happens to the temperature and salinity distribution also. Were such processes considered, and how would they affect the surface height?

Reply. Yes, we would also expect some changes in T,S fields and their potential effect on free sea level oscillations of seasonal scale. Indeed, considering that these oscillations occur only in baroclinic conditions, they can be related to spatial variability of the T and S (e.g. water density). However, an investigation of this interesting problem was out of scope of presented manuscript and will be studied in future.

l. 499-502. Most of the interannual variability in the seasonal sea level fluctuations likely originates directly from the interannual variability in the atmospheric forcing. E.g. the role of the air pressure conditions, the NAO index, etc., have been shown to explain a significant portion of the interannual variability. Thus, I suppose the contribution from the baroclinic free oscillations is minor. (Which might be mentioned.) L. 499-502.

Reply: We agree that the interannual variability in the seasonal sea level fluctuations is apparently related to interannual variability of seasonal fluctuations of the wind and atmospheric pressure. This fact was supported by many researches (Ekman and Stigebrandt, 1990; Ekman , 1998; Plag and Tsimplis, 1999; Stramska et al, 2013; Barbosa and Donner, 2016; Cheng et al, 2018). This is also true that our estimated amplitudes of free baroclinic oscillations of seasonal scale are low (2,5 – 5,5 cm). Nevertheless, these amplitudes are of the same order as the amplitudes of annual Baltic Sea level variability (4 - 13 cm) estimated using stationary approximation from the tide gauge observations for 60 year period (Ekman, 1996). Corresponding phrase

was added to the Discussion section. Indeed, for non-stationary process observed annual amplitudes are higher and can reach 30-40 cm (Ekman and Stigebrandt, 1990; Medvedev, 2014). In our Discussion section we put forward the idea that the role of found in our study baroclinic free oscillations under combination of specific conditions (resonance, favorable stratification), of cause, occurring not each year, might be non-negligible. And this question requires more clarification in future studies.

Please also note the supplement to this comment:
https://os.copernicus.org/preprints/os-2020-110/os-2020-110-AC2-supplement.pdf

———————————————————

[Figure]

[Figure]

Figure XX. Amplitude spectra A(ω) of tide gauges at Helsinki station for 2009-2010 (grey line) and of simulated free oscillations in barotropic (a) and baroclinic (b) conditions (red line).

**Fig. 1.** Figure XX

**Supplement:**

We appreciate the Referee valuable remarks, recommendations and his profound corrections and carefully addressed them in the new version of the manuscript. . Our answers on the major comments can be found in the text bellow and full answers are given in the Supplement materials.

On behalf of all authors,

Elena Zakharova

Referee 2 major comment

The oscillations studied in this paper occurred in a model where some parameters were adjusted to unrealistic values in order to reduce damping (l. 182-183). It would be interesting to see some discussion on how the results obtained relate to the sea level behaviour in the real Baltic Sea.

How much are such oscillations expected to contribute to the real sea level variability? Is there a possibility that the parameter adjustments affect the oscillation frequencies?

**Reply:**

**1.The parameters of adjustments affect the dissipation speed and do not affect the frequencies. If we do not reduce these parameters we can't catch the oscillations of seasonal scale.** *We edited corresponding phrase in the text for clarification.*

**2. A comparison of tide gauge and numerical simulation spectra is shown on figure XX . The real sea level fluctuation is the superposition of forced and free oscillations of different origin. The tide gauge spectrum contains very big amount of peaks those amplitudes are significantly higher.  In contrast, the spectrum of simulated free oscillations is characterised by a small number of peaks of lower amplitudes which are masked by forced oscillations. So, we decided to exclude comparison of real and simulated spectra from the discussion.**

[Figure]

**Figure XX. Amplitude spectra A($\omega$) of tide gauges at Helsinki station for 2009-2010 (grey line) and of simulated free oscillations in barotropic (a) and baroclinic (b) conditions (red line).**

How fast would they be damped?

**Reply: The answer for this question is given in the Section 2.3. "Under natural conditions, the free sea level oscillations attenuate rapidly due to the dissipative effects of vertical and horizontal viscosity, near-bottom friction, non-linear effects, and Earth's rotation (Proshutinsky 1993, Zakharchuk et al., 2004). According to a theoretical concept and previous numerical experiments (Proudman, 1953; Wübber and Kraus, 1979; Zakharchuk et al., 2004; Leppäranta and Myrberg, 2009), the relaxation of the Baltic large-scale free sea level oscillations takes several days."**

Fig. 8: Why is there so much white space in these maps? The areas around the oscillation nodes are apparently excluded due to low amplitude. But why are e.g. phase speeds for the eastern Gulf of Finland missing in Fig. 8b, even if the amplitude of the oscillation should be high (Fig. 7b)?

**Reply: We introduced additional phrase to the caption of corresponding figures and following phrase into the body text. "In areas where $\Delta F_x$ and $\Delta F_y$ equal to zero (white areas on fig 8), the standing wave component prevails."**

In a seasonal scale in the baroclinic simulation, after all the external forcing ceases, I would assume that something happens to the temperature and salinity distribution also. Were such processes considered, and how would they affect the surface height?

**Reply. Yes, we would also expect some changes in T,S fields and their potential effect on free sea level oscillations of seasonal scale. Indeed, considering that these oscillations occur only in baroclinic conditions, they can be related to spatial variability of the T and S (e.g. water density). However, an investigation of this interesting problem was out of scope of presented manuscript and will be studied in future.**

l. 499-502. Most of the interannual variability in the seasonal sea level fluctuations likely originates directly from the interannual variability in the atmospheric forcing. E.g. the role of the air pressure conditions, the NAO index, etc., have been shown to explain a significant portion of the interannual variability. Thus, I suppose the contribution from the baroclinic free oscillations is minor. (Which might be mentioned.)

L. 499-502.
**We agree that the interannual variability in the seasonal sea level fluctuations is apparently related to interannual variability of seasonal fluctuations of the wind and atmospheric pressure. This fact was supported by many researches (Ekman and Stigebrandt, 1990; Ekman , 1998; Plag and Tsimplis, 1999; Stramska et al, 2013; Barbosa and Donner, 2016; Cheng et al, 2018). This is also true that our estimated amplitudes of free baroclinic oscillations of seasonal scale are low (2,5 – 5,5 cm). Nevertheless, these amplitudes are of the same order as the amplitudes of annual Baltic Sea level variability (4 - 13 cm) estimated using stationary approximation from the tide gauge observations for 60 year period (Ekman, 1996). *Corresponding phrase was added to the Discussion section*. Indeed, for non-stationary process observed annual amplitudes are higher and can reach 30-40 cm (Ekman and Stigebrandt, 1990; Medvedev, 2014). In our Discussion section we put forward the idea that the role of found in our study baroclinic free oscillations under combination of specific conditions (resonance, favorable stratification), of cause, occurring not each year, might be non-negligible. And this question requires more clarification in future studies.**

======================================================================

l. 144-145: sigma_p is not the relation of sigma_m and sigma_tg (as stated), but sigma_er and sigma_tg.
**The correction was introduced."The standard deviation of the observed ($\sigma_{tg}$) and simulated ($\sigma_m$) sea surface height.."**

l. 145-146: Please give the definition of P_m. Now it remains unclear which measure should be <0.674*sigma_tg.
**The phrase was corrected. "The $P_m$ parameter allows the assessment of the NUMBER of good simulations (comparing to total number of outputs) considering their accuracy < 0.674$\sigma_{tg}$."**

l. 201: ...a_k and b_k are the coefficients...
**The correction was introduced.**

l. 202: "mean average" => "mean" or "average"
**The correction was introduced.**

l. 207-211: Is the period P same as T above? If so, please use the same symbol. If not, please explain.
**The correction was introduced: P was changed for T in the text and corresponding equations.**

Fig. 4: The original amplitude of the displacement ranges from -50 to +100 cm in Fig.3. How does this relate to the plots in Fig. 4 starting from around +10-20 cm at every station? What is the vertical axis in these?

**The Fig 3b shows the sea level under the meteorological forcing, while on the Fig4 we demonstrate the simulated FREE se level oscillations AFTER cessation of the forcing. For more details on comparison of free and real oscillations see the figure XX above.**

Fig. 4: It would ease the comparison of amplitudes if all subplots had the same y axis (as they are very close to each other already).
**The correction was introduced. The Y-axis scale is unified.**

l. 229 and Fig. 5: What is "standard deviation of amplitudes"? If this is the standard deviation of time series, as Eq. (4) implies, then the word "amplitude" here is misleading.
**The title was changed for "Sea level standard deviation of free barotropic oscillations.."**

l. 230: Please check the font size of all subscripts, here and elsewhere.
**The correction was introduced.**

l. 233: Consider adding locations of Pärnu Bay and Rügen Island to Fig. 1.
**The map on the Figure 1 was updated.**

l. 235-236: "Oscillations of medium intensity can be noted as over local uplifts in the Baltic Proper as over-bottom depressions..." Please reformulate this sentence.
**The sentence was rewritten. " Oscillations of medium intensity occur in area of local uplifts in the Baltic proper, in area of the Ulvö Deep, the Landsort Deep, the Northern Deep and the Gotland Deep."**

l. 261: There is a 17-h period listed. However, there are no 17-h peaks in Fig. 6. This should be 16-h?
**Thank you for catching this error. The text was edited. "In our experiment, the corresponding periods were 31, 27, 23, 20, 16, and 13 hours "**

l. 310: Leppäranta and Myrberg, 2009
**The correction was introduced.**

l. 311: From Eq. (13), a range of 12-67 m/s corresponds to depths of 15-458 m. From where does the lowest limit of 15 m come? Please specify.

**Again, thank you for your vigilance. We verified data used against data of (Lepparana and Myrberg, 2009). The correct range of mean depths in studied areas are 15-77 m and the range of maximal depth is 14–458 m. Necessary corrections were introduced**

l. 321: "anemobaric forcing": please specify what is meant by this. Is "anemobaric" a synonym to the entire meteorological forcing described on lines 118-119?
**Anemobaric forcing was changed for meteorological forcing.**

l. 322: "30-35 cm in amplitude" => "in range"? In Fig. 9, the largest range of variations seems to be 35 cm. This is not amplitude, which by definition is half of the total range.
**The correction was introduced: "30-35 cm in range"**

l. 330: "standard deviation of the amplitudes"; see comment above.
**Correction was made.**

l. 336: Ellesmere Island? Please check the name, and add location to Fig. 1.
**We meant Öland Island area. Correction was introduced.**

l. 362: "Free oscillations of 27 h periods in the baroclinic conditions reached the maximum in the narrow zone near the southwest Finland coast". There seems to be a much more apparent maximum in the eastern Gulf of Finland in Fig. 12b, please check this sentence.
**Thank you, we agree with remark. The corresponding changes were made.**

l. 382-383: "Dash line on the histogram plots indicates minimum theoretical value of phase speed of baroclinic (Ci) and barotropic (Cg) gravity waves." It looks like the dash line for Ci indicates the maximum value (1.53 m/s), not minimum.
**We agree with remark. The corresponding changes were made.**

l. 412: May be more specific here: "significantly lower for 358-day waves and belong to the theoretical range for 89-day waves". I see this is what is meant, but "longer" and "shorter" are a bit too generic and it is hard to understand the sentence.
**The text was rewritten and, we hope, now is more simple and clear. " For 358-day waves, our estimations of the phase speed are significantly lower than that of the theoretical internal gravity waves ($C_i$). For 89-day waves, the part of our estimations belongs to the range of phase speed of internal gravity waves."**

Fig. 15. Line 388 says the Ci range is 0.08-1.53 m/s. Why is the maximum lower here?
**Thank you for this remark. The figure was redrawn. The line corresponding to max value 1.53 m/s was removed as it is out of x-axis limits. The XLim was left as previously as at a larger range the details of the histogram will be invisible.**

l. 467: decreases => increases
**Thank you for catching this error. The correction was introduced.**

l. 479: It would be helpful to mention explicitly the period of inertial oscillations in the area (about 14 hours).
**The correction was introduced.**